# Analysis of an Extreme Weather Event in a Hyper-Arid Region Using WRF-Hydro Coupling, Station, and Satellite data

Youssef Wehbe[1,2], Marouane Temimi[1], Michael Weston[1], Naira Chaouch[3], Oliver Branch[4], Thomas Schwitalla[4], Volker Wulfmeyer[4], Xiwu Zhan[5], Jicheng Liu[5,6], and Abdulla Al Mandous[2]

[1] Department of Civil Infrastructure and Environmental Engineering, Khalifa University of Science and Technology, Masdar City, P.O. Box 54224 Abu Dhabi, United Arab Emirates
[2] National Center of Meteorology (NCM), P.O. Box 4815, Abu Dhabi, United Arab Emirates
[3] NOAA CREST Institute/City University of New York, New York, NY, United States
[4] Institute of Physics and Meteorology, University of Hohenheim, Garbenstraße 30, D-70599 Stuttgart, Germany
[5] NOAA-NESDIS Center for Satellite Applications and Research (STAR) NOAA
[6] ESSIC/CICS, University of Maryland College Park, College Park 20740, MD, USA

*Correspondence to*: Youssef Wehbe (ywehbe@ncms.ae)

**Abstract.** This study investigates an extreme weather event that impacted the United Arab Emirates (UAE) in March 2016 using the Weather Research and Forecasting (WRF) model version 3.7.1 coupled with its hydrological modeling extension package (Hydro). Six-hourly forecasted forcing records at 0.5° spatial resolution, obtained from the NCEP Global Forecast System (GFS), are used to drive the three nested downscaling domains of both standalone WRF and coupled WRF/WRF-Hydro configurations for the recent flood-triggering storm. Ground and satellite observations over the UAE are employed to validate the model results. The model performance was assessed using precipitation from GPM (30-minute, 0.1° product), soil moisture from AMSR2 (daily, 0.1° product) and NOAA SMOPS global product (6-hourly, 0.25° product), and cloud fraction retrievals from MODIS (daily, 5 km product). The Pearson correlation coefficient (PCC), relative bias (rBIAS) and root-mean-square error (RMSE) are used as performance measures. Results show reductions of 24% and 13% in RMSE and rBIAS measures, respectively, in precipitation forecasts from the coupled WRF/WRF-Hydro model configuration, when compared to standalone WRF. The coupled system also shows improvements in global radiation forecasts, with reductions of 45% and 12% for RMSE and rBIAS, respectively. Moreover, WRF-Hydro was able to simulate the spatial distribution of soil moisture reasonably well across the study domain when compared to AMSR2 soil moisture estimates, despite a noticeable dry/wet bias in areas where soil moisture is high/low. Temporal and spatial variability of simulated soil moisture compare well to estimates from the NOAA SMOPS product, which indicates the model capability to simulate surface drainage. Finally, the coupled model showed a shallower PBL compared to the standalone WRF simulation, which is attributed to the effect of soil moisture feedback. The demonstrated improvement, at the local scale, implies that WRF-Hydro coupling may enhance hydrological and meteorological forecasts in hyper-arid environments.

## 1 Introduction

Changes in rainfall patterns directly impact hydrological processes overall and particularly the timing and magnitude of floods. In order to produce reliable flash flood warnings, accurate predictions of precipitation timings and amounts, along with its impact on resulting runoff, are needed. However, discrepancies are largely reported when forecasting precipitation using numerical weather prediction (NWP) models. The inaccurate predictions are then magnified in flood forecasts when simulated rainfall is used to drive a hydrological model (Wang and Seaman, 1997;Nielsen-Gammon et al., 2005;Yousef and Ouarda, 2015). Standard hydrological models are often driven by precipitation products inferred from radar, rain gauge, and remote sensing observations or a combination of them. The lack of dense radar and rain gauge networks in areas like the Arabian Peninsula makes the reliance on precipitation remote sensing more attractive. However, such products come with coarse spatial resolution and fail to capture rainfall structures forced by mesoscale orography and land surface interactions, which are prevalent across the Arabian Peninsula, including the UAE (Mandoos, 2006). A number of studies reported higher inaccuracies of precipitation products in arid regions (Fekete et al., 2004;Milewski et al., 2015;Wehbe et al., 2017;Wehbe et al., 2018), which suggests the interest in enhancing mesoscale modeling of weather processes in such regions to generate reliable precipitation products and therefore accurate prediction of extreme hydrometeorological events. The Weather Research and Forecasting (WRF) model is a mesoscale NWP system created for the dual purpose of assisting with the needs of operational forecasting and facilitating atmospheric research. WRF is designed to be a next-generation mesoscale forecast model and data assimilation system, for the purpose of advancing the understanding and prediction of mesoscale weather and accelerating the transfer of research advances into operations (Skamarock et al., 2005). Powers et al. (2017) give a detailed overview of the initial phases of the WRF project.

A limited number of studies utilizing the WRF model focused on areas in the Middle East region. Awad et al. (2007) summarize the achievements of the UAE Air Force and Air Defense in the use of the WRF model, where a local operational suite was developed for predicting weather numerically over the Middle East region generally, and the Arabian Peninsula and UAE areas specifically. Recently, Chaouch et al. (2017) studied the consistency of WRF simulations with seven different planetary boundary layer (PBL) schemes and showed that better performance is obtained with the Quasi-Normal Scale Elimination (QNSE-EDMF) scheme while the remaining schemes showed comparable performance. El Afandi et al. (2013) simulated heavy rainfall events in the Sinai Peninsula using WRF, for the purpose of exploring how early warnings could be issued for flash flood risk mitigation. They found WRF simulations of a flash flood event on January 18, 2010 consistent with measurements recorded at rainfall gauges for different parts of the Peninsula with RMSE values below 5%.

The enhanced WRF Hydrological modeling extension package (WRF-Hydro) has shown improvement in prediction capabilities of hydrometeorological forecasts using numerical prediction tools when tested over other regions with climate conditions different from those observed in the Middle East (Parodi et al., 2013). It has been used for flash flood prediction, land-atmosphere coupling studies, regional hydroclimate impacts assessment, and seasonal forecasting of water resources (Gochis et al., 2013b). Flash flood predictions using the WRF-Hydro model have been applied across the United States (Unal,

2015;Gochis et al., 2015;Read, 2015) and various parts of the world. Fiori et al. (2014) analysed a convective system responsible for an extreme flash flood event that occurred in Genoa, Italy on November 4, 2011, using WRF coupled with WRF-Hydro. The study outlined the effectiveness of the model in predicting quantitative precipitation, for the purpose of flash flood forecasting. Streamflow forecasting from the fully coupled WRF-Hydro modeling system was evaluated through comparisons with both observations and uncoupled hydrological model results. The results of the study highlighted the need to consider multiple factors and sources of error for the prediction of hydrometeorological events, and presented optimal configurations of WRF-Hydro for future extreme flash flood events in the Mediterranean region. A study conducted for the western Black Sea region of northern Turkey by Yucel et al. (2015) also analysed the potential of the WRF-Hydro modeling system for flash flood predictions. The study explored the potential of improving runoff and streamflow simulations through the application of data assimilation for precipitation prediction in NWPs. The involved analyses concentrated on assessing the capabilities of the WRF and WRF-Hydro models in forecasting the responses of floods for the vast range of hydrological settings associated with 10 different events that occurred in the study area. Precipitation inputs for the forcing of the WRF-Hydro model were derived from the WRF model and the EUMETSAT Multi-sensor Precipitation Estimates (MPEs). The use of a data assimilation scheme with the WRF model provided significant improvements in simulations of streamflow, while the MPE product led to less accurate streamflow simulations. The optimum results in error reduction were obtained when both WRF model data assimilation and hydrological model calibration were employed.

In a recent study, Givati et al. (2016) calibrated and evaluated the coupled WRF/WRF-Hydro modeling system with the standalone WRF model for flood forecasting of Wadi Musrara, Israel, for two major winter storm events in January and December 2013. Higher correlations (0.89 and 0.85) with the station observations were recorded by the coupled WRF-Hydro model than those of the standalone WRF model (0.85 and 0.80) for both events. Lower RMSE values were also obtained by the coupling (12.2 and 24 compared to 16 and 30). The findings of the study showed that the coupled WRF/WRF-Hydro modeling system resulted in improved precipitation and hydrological simulations when compared with the results of the standalone WRF simulations. Therefore, the authors proposed the employment of atmospheric-hydrological coupling due to its potential to produce improved precipitation predictions, which translates to better hydrological forecasts for early flood hazard mitigation. The application of WRF-Hydro for flood forecasting in arid environments has also been has also been explored by Silver et al. (2017). They simulated six storm events over seven basins in arid and semi-arid regions of Israel and Jordan using WRF-Hydro, while ingesting field-based soil characterization data into the land surface model initialization. Nash–Sutcliffe efficiency coefficient (NSE) values of up to 0.415 were recorded between the observed streamflow records and WRF-Hydro simulated streamflow. In desert regions, high soil porosity and hydraulic conductivity of the prevailing sandy soil implies rapid infiltration and runoff drainage. This suggests that the impact on latent heat and, therefore, on the surface radiation budget would be minimal. Alternatively, precipitation largely contributes to soil physical crust formation in desert environments (Fang et al., 2007). Precipitation compacts fine particulate and fills the porosities of the top soil layer, forming a hard shell. Dust is also washed out of the atmosphere by precipitation over desert environments which increases amounts of finer particulate at the surface layer to further accelerate crust formation process. This translates to less vertical infiltration and

more lateral flow processes. These mechanisms are specific to arid regions and corroborate the importance of accounting for lateral flow and surface feedback in the coupled WRF/WRF-Hydro model to correctly capture the atmospheric and hydrological process.

The present study expands the ongoing research addressing the prediction of extreme hydrometeorological events in arid regions and investigates the potential of coupling atmospheric and hydrological processes in short-term prediction. There is interest in determining the potential improvement in the simulation that the online coupling of atmospheric and hydrological could bring in the case of an extreme hydrometeorological event. To our knowledge, such coupling has never been assessed in hyperarid environments like the one observed in the UAE where hydrological and atmospheric processes are specific and different from other study domains where similar coupling was evaluated. We focus in this study on an extreme hydrometeorological event recorded on March 9, 2016 over the UAE. The standalone WRF and fully coupled WRF/WRF-Hydro simulations are assessed against (i) weather station surface observations of rainfall, 2-meter surface temperature, and global radiation and (ii) satellite remote sensing retrievals of GPM rainfall, AMSR2 soil moisture, and MODIS/Terra cloud fraction. In the absence of ground-based streamflow gauges in the study domain, satellite data were valuable to understand the dynamics of the event and changes in precipitation and soil moisture distributions.

# 2 Study Domain and Datasets

## 2.1 Case Study

This study consists of three nested domains (see Fig. 1), with the parent domain covering the Arabian Peninsula, Iran, Iraq, Afghanistan, and parts of Pakistan, Syria, and Ethiopia (9° to 37 °N and 39° to 70 °E), thus capturing a broad range of weather systems. The UAE (22° to 27° N) is part of the arid climatic regime of the Arabian Peninsula, characterized by high temperatures and dry environment during summers and mild wet winters. The region is located in two distinct climate zones: subtropical north of 20°N and the tropical/monsoonal southern part (AlSarmi and Washington, 2011;Almazroui et al., 2012). Winter and early spring occur from January to April. During this period the region is affected by the Siberian High pressure with cold air mass and the Red Sea trough (RST) in the northern and southern parts of the Peninsula, respectively (Tsvieli and Zangvil, 2007). The summer season extends from June to August and is greatly impacted by the Indian Monsoon depression. The Monsoon is associated with the Low-Level Jet (LLJ) affecting the southern part of the Peninsula (Sathiyamoorthy et al., 2013). The Tropic of Cancer, located under the descending limb of the Hadley Cell, leads to the region being dominated by subtropical anticyclones, associated with air subsidence, stable conditions, and high pressure (Mandoos, 2006).

Fig. 1 also shows the high resolution (100-meter) WRF-Hydro terrain model domain over the UAE, along with the associated topography map derived from the 30-meter resolution Advanced Spaceborne Thermal Emission and Reflection Radiometer (ASTER) digital elevation model (DEM) (Toutin, 2008). The Al Hajar Mountains, located along the Gulf and Oman coasts, foster local convergence zones that trigger small-scale convection initiation (Chaouch et al., 2017). Consequently, the northeastern part of the UAE receives more rainfall compared to the country’s 100 mm annual average (Ghebreyesus et al.,

2016a, b;Ouarda et al., 2014;Wehbe et al., 2017). The inland city of Al Ain is of close proximity to the Al Hajar Mountains on the southerly downside, designating it as a susceptible area to flash floods triggered by accumulated upstream runoff from the northern highlands. Farming and agriculture is concentrated in this area to benefit from the oasis effects of high rainfall rates and fresh groundwater, compared to other parts of the country.

On March 9, 2016, a low pressure system passing from the UAE and Oman to southeastern Iran produced thunderstorms, fierce winds, large hail, and severe flooding. According to the UAE National Center of Meteorology (NCM), over 240 mm of rain was recorded in Dubai while winds of up to 126 km/h battered the capital (Adonai, 2016). The movement of an Active Red Sea Trough, associated with hot and dry weather resulting from east-southeasterly flows in the lower troposphere, and accompanied by a cold upper-tropospheric trough extending from the north, resulted in unstable stratification. Such stratification resulted in the development of a mesoscale convective system. The accumulation of clouds over the Western side of the UAE developed and gradually moved towards the coastal areas. The influence of the south-easterly moist air contributed to the evolution of the clouds, so that their vertical extent exceeded 5 km, according to radiosondes retrieved by the NCM at Abu Dhabi Airport. The resulting skew-T profiles are shown in Fig. 2 for 09/03/2016 at 00Z (a) and 12Z (b). The temperature inversions in the surface layer (80 – 761 m) depicted in Fig. 2a indicate the onset of the event with convective available potential energy (CAPE) values reaching 506.3. Fig. 2b shows the extent of the towering cumulus clouds from cloud bases of 770 m (920 hPa) at the lifting condensation level (LCL) to cloud tops at the divergence of the overlapping dew point and temperature profiles at the 6200 m (530 hPa) level. The freezing level triggering the heavy rain event was reached above 4100 m.

## 2.2 Datasets

Satellite and ground-based observations were used for the verification of WRF and WRF-Hydro simulations. In the case of the former, necessary data re-gridding was done to match WRF outputs spatially and temporally. The following datasets were used to quantitatively and qualitatively assess the performance of the simulations.

**<u>The Global Precipitation Measurement (GPM)</u>:**

The simulated precipitation was compared to GPM satellite retrievals. The GPM mission, launched in February 2014, provides higher temporal (30-minute) and spatial (0.1°) resolution precipitation estimates through the Integrated Multi-satellitE Retrievals (IMERG) product. The GPM IMERG product inter-calibrates, merges, and interpolates GPM constellation satellite microwave precipitation estimates with microwave-calibrated infrared estimates, and rain gauge analyses to produce a higher resolution and more accurate product (Huffman et al., 2014). The GPM core satellite estimates precipitation from two instruments, the GPM Microwave Imager (GMI) and the Dual-Frequency Precipitation Radar (DPR). More importantly for this study, the GPM radar has been upgraded to two frequencies, adding sensitivity to light precipitation.

**The Advanced Microwave Scanning Radiometer 2 (AMSR2):**

Soil moisture estimates from the AMSR2 mission were used to verify the simulated soil moisture and its simulated spatial extent. In this study we rely on AMSR2 datasets for the verification of simulated soil moisture as products from other datasets were masked out over the study area during the investigated event. AMSR2 was launched on May 18, 2012 on board the Global Change Observation Mission 1$^{st}$-Water (GCOM-W) platform, with the capability of measuring passive microwave emissions from the surface and atmosphere. The window channels on board of the sensor are capable of retrieving key surface parameters like soil moisture. AMSR2 L3 datasets provide daytime and night-time soil moisture measurements with near-global coverage over 3 days (Wentz et al., 2014). Moreover, AMSR2 is equipped with frequencies higher than the L band that are more suitable for soil moisture retrievals. Such frequencies, ranging from 6.93 – 89 GHz, are currently available on board Soil Moisture and Ocean Salinity (SMOS) and Soil Moisture Active Passive (SMAP) missions.

**The MODIS/Terra Joint Cloud, Aerosol, Water Vapour and Profile (MODATML2):**

The MODATML2 dataset from Platnick et al. (2015) contains measurements in 36 spectral bands (250 to 1000 m resolution in nadir) for a combination of key atmospheric parameters at daily and 5 or 10-km resolutions (parameter dependent), including aerosol properties, water vapor profiles, and cloud properties, starting October 13, 2003 and ongoing. The cloud mask is derived as a probabilistic variable from multispectral testing techniques proposed by Ackerman et al. (1998). The cloud fraction (10-km resolution), used in this study, is obtained from the Terra platform's infrared retrievals during both day and night.

**Weather Stations:**

Rainfall, surface temperature (2-meter), and average global radiation records were obtained from a network of 57 automatic weather stations (see Fig. 1) operated by the NCM across the UAE. Rainfall observations were recorded at 15-minute intervals during the event, while hourly observations of temperature and global radiation were made available after quality control.

## 3 Methodology

Two simulations were carried out in this study; one with the standalone WRF version and another with the online coupled WRF/WRF-Hydro version. In both simulations, the initial and lateral boundary conditions of the parent domain were defined from the Global Operational Analysis and Forecast Products of the National Center for Environmental Prediction (NCEP-GFS) at 0.5° spatial resolution and 6-hour intervals (00, 06, 12, 18 UTC). The static terrain attributes and topography used in the WRF pre-processing system (WPS) were derived from United States Geologic Survey (USGS) datasets (Smart et al., 2005). A description of the configuration of each version is in the following sections.

The existing Noah-Multiparameterization Land Surface Model (Noah-MP LSM) in standalone WRF considers single vertical columns (one-dimensional) of terrain properties at each overland grid cell (Niu et al., 2011;Ek et al., 2003). It fails to account for horizontal interactions between adjacent grid cells to calculate soil moisture, temperature profiles, runoff, and water and

energy fluxes at the land surface. Therefore, the runoff–infiltration partitioning in the standalone WRF simulation is described as a purely isolated vertical process with no intake from (or discharge to) neighbouring grid cells, as dictated by topography. On the other hand, the WRF-Hydro model utilizes the Noah-MP LSM 1-D representations and attempts to improve the simulation of terrestrial hydrologic processes at high spatial and temporal resolutions by including lateral redistribution of overland and saturated subsurface flows for runoff prediction (Gochis et al., 2013c). More importantly for this study, WRF-Hydro is run in coupled mode with WRF to permit the feedback of land surface fluxes of energy and moisture to the atmosphere, which impacts the simulated precipitation fields (Arnault et al., 2016;Larsen et al., 2016;Senatore et al., 2015;Koster et al., 2004). Hence, the coupled WRF/WRF-Hydro configuration differs from the standalone WRF configuration by its (i) lateral distribution of surface runoff and (ii) feedback of surface fluxes to the atmosphere.

### 3.1 Standalone WRF

The model configuration and 1:3 downscaling ratio was setup as recommended by Givati et al. (2011), consisting of three nested domains – d01 (parent domain) with a 350 × 350 grid and 9 km resolution; d02 (intermediate domain) with a 403 × 403 grid of 3 km resolution; and d03 (inner domain) with a 562 × 562 grid at 1 km spacing (see Fig. 1).

Based on the comparable performance of PBL schemes over the UAE (Chaouch et al., 2017) and given the unstable conditions during the event, the non-local YSU scheme is selected as the most favourable scheme for maintaining entrainment flux proportional to surface flux (Hariprasad et al., 2014;Hu et al., 2010;Shin and Hong, 2011). The selection of a microphysics scheme has been shown to be of least importance compared to PBL and cumulus schemes (Argüeso et al., 2011).

In light of the findings of Givati et al. (2011), the physics options chosen for this study included: the Noah-MP land surface scheme (Niu, 2011), Monin–Obukhov surface layer scheme (Monin and Obukhov, 1954), Rapid Radiative Transfer Model for General Circulation Models (RRTMG) longwave (Mlawer et al., 1997) and shortwave radiation schemes (Iacono et al., 2008), and the Morrison double-moment cloud microphysics scheme (Morrison et al., 2009) with improved strati formed cloud persistence compared to the single-moment scheme. The WRF and WRF-Hydro configurations are listed in Table 1.

### 3.2. Coupled WRF/WRF-Hydro

The WRF Hydrological modeling extension package (WRF-Hydro) from Gochis et al. (2013a), was developed through research collaborations between the National Center for Atmospheric Research (NCAR) and its partners, the National Aeronautics and Space Administration (NASA) and the National Oceanic and Atmospheric Administration (NOAA). It can be used as a standalone model and has an architecture to facilitate coupling between hydrological models and atmospheric models. It provides the capability to spatially relate meteorological variables to physical and terrestrial attributes (elevation, soil, land use, etc.) and their associated feedbacks. Several hydrologic routing physics options exist in the version 3.0 of WRF-Hydro. Here, a fully distributed, 3-dimensional overland surface flow model configuration was employed. Soil moisture initialization was provided from the WPS along with the predefined soil hydraulic parameters for the Noah-MP LSM on the basis of USGS soil classifications (Ek et al., 2003).

The GIS Python-based pre-processing approach from Gochis and Sampson (2015) was followed to derive the WRF-Hydro LSM and routing grids. Inputs of the static terrain properties from the WPS GEOGRID file and the high-resolution (30 m) ASTER DEM were used. A re-gridding factor of 10 was applied to reach the 100 m LSM resolution from the 1 km GEOGRID resolution. The minimum basin size was defined by a threshold of 20 pixels per stream, following the analytical method for stream network delineation proposed by Tarboton et al. (1991). Overall, a dominant parallel pattern of streams network is obtained, draining water westward from the mountainous region in eastern UAE towards the Gulf on the west side (Fig. 1). The northern part of the UAE shows a large number of streams flowing in the opposite direction towards the east side of the country. The sandy nature of the soil in the desert in the southwestern side of the country favours rapid infiltration of runoff and its drainage towards the Gulf as groundwater, before it makes surface around the western coastal regions (Fig. 1). This contributes to the existence of salt flats around those regions. To our knowledge, a dense drainage network which may allow us to identify the main wadis in the western region does not exist in the UAE and this study proposes a first version of such network. The challenge faced when studying hydrological processes in hyperarid regions, like the UAE, is the absence of gauged watersheds and lack of in situ data to calibrate and verify the hydrological models. In this study, the performance of the hydrological simulations was evaluated using satellite soil moisture data and the analysis of the changes in its lateral extent.

### 3.3 Statistical Performance Measures

The RMSE and relative BIAS methods of evaluation were implemented, with the observed value representing the station measurement, or the satellite retrieval in the absence of a station observation (Eq. 1 and 2). The RMSE and rBIAS reflect the average error and degree of over- or underestimation of the model output fields.

$$\mathrm{RMSE} = \sqrt{\frac{\sum_{i=1}^{n}(y_{o\,i} - y_{est\,i})^2}{n}}, \tag{1}$$

$$\mathrm{rBIAS} = \frac{\sum_{i=1}^{n}(y_{est\,i} - y_{o\,i})}{\sum_{i=1}^{n} y_{o\,i}} \times 100, \tag{2}$$

where, $y_{est\,i}$ and $y_{o\,i}$ are the estimated (simulated) and observed values, respectively, at station i, and n is the number of observations.

The Pearson correlation coefficient (PCC) (Benesty et al., 2009) was used in relating the observed values and model output (Eq. 3). The Pearson correlation reflects the statistical association between variables, and can range between -1 to 1, where 1 is total positive linear correlation, 0 is no linear correlation, and −1 is total negative linear correlation.

$$\mathrm{PCC} = \frac{\sum_{i=1}^{n}(y_{o\,i} - \bar{y}_o)(y_{est\,i} - \bar{y}_{est})}{\sqrt{\sum_{i=1}^{n}(y_{o\,i} - \bar{y}_o)^2 \times \sum_{i=1}^{n}(y_{est\,i} - \bar{y}_{est})^2}}, \tag{3}$$

# 4 Results and Discussion

## 4.1 Analyses of Atmospheric Variables

### 4.1.1 Gauge Rainfall versus GPM, WRF, and WRF/WRF-Hydro

Fig. 3 shows the cumulative event rainfall over the UAE in d03 between 06Z 08/03/2016 and 06Z 10/03/2016. This includes values recorded by the rain gauges (a), retrieved from the GPM product (b), and simulated from the WRF and WRF/WRF-Hydro runs (c and d, respectively). The highest cumulative rainfall of 288 mm was recorded by the rain gauge at Al Shiweb weather station (see Fig. 3a), in the northeastern part of the UAE. Lower rainfall rates between 30 to 70 mm were recorded along the coast and western parts of the country. The minimum, maximum, and mean from each rainfall source are listed in Table 2, in addition to the correlation, RMSE, and rBIAS values of the simulated and GPM-retrieved rainfalls against the station records over the UAE.

GPM retrievals recorded a maximum cumulative rainfall of 189 mm over the eastern coast of Oman which is lower than the maximum obtained from the station data in the UAE (readings in Oman are not available), which implies that GPM could have underestimated rainfall records as it did not capture the maximum cumulative reading reported at Al Shiweb weather station (288 mm). This could be explained by the lower temporal coverage of GPM (30 minutes) compared to the stations (15 minutes) which could have led to the former missing the peak of rainfall records. Another reason that could explain the underestimation is the coarse spatial resolution of GPM data (10 km), which senses rainfall over a footprint larger than the one represented by the local rain gauge readings. On the other hand, GPM overestimates cumulative rainfall values ranging between 50 to 90 mm in the northeastern part of the UAE and highlands (see Fig. 3b). Minimum rainfalls between 10 to 40 mm were recorded by stations along the western coast lines and central inlands, while no significant rainfall was recorded over the western areas. In a study that validated a number of precipitation products over UAE for a 10-year period (Wehbe et al., 2017), it was shown that the analysed remote sensing products perform better at higher elevations (>250 m), which is the case of Al Shiweb weather station (306 m). In line with the previous study, the lowland stations in the western region showed higher biases by GPM and retrievals over areas not receiving any rainfall during the event. More events are needed to accurately assess the performance of GPM in the region and demonstrate any potential improvement over its predecessor, TRMM – Tropical Rainfall Measurement Mission (Huffman et al., 2007), among others.

The standalone WRF and fully coupled WRF/WRF-Hydro simulated rainfall exhibited spatial rainfall patterns similar to those obtained from GPM and station data, with the highest rainfall simulated over the coast of Oman and eastern UAE (see Fig. 3c and 3d). However, the coupled WRF/WRF-Hydro model resulted in a maximum cumulative rainfall of 156 mm over the northeastern border with Oman, which is closer to the 189 mm obtained from GPM readings compared to the 122 mm from the standalone WRF system. Also, more rainfall was reproduced over the western quarter by the standalone WRF than the coupled WRF/WRF-Hydro where GPM and stations do not report significant readings. Overall, more of the rainfall retrieved by GPM or measured by the stations was reproduced by the coupled WRF/WRF-Hydro model and a better agreement in the spatial pattern of rainfall and its magnitude is obtained.

More importantly, the lowest rBIAS was obtained from the WRF/WRF-Hydro setup at 0.21, followed by the WRF setup (0.34), and the GPM retrievals (0.71). The higher bias associated with the GPM rainfall retrievals is thought to be related to the difference in spatial scales, ice-scattering microwave retrieval mechanism, locality of orographic rainfall events, among other factors. The improved performance of the coupled WRF/WRF-Hydro system compared to the standalone WRF is evident across all measures in Table 2 – closer to the observed 48 mm mean and 288 mm maximum (39 vs. 32 mm and 156 vs. 122 mm), higher correlation (0.82 vs. 0.76), and lower RMSE (10.89 vs. 14.24) and rBIAS (0.21 vs. 0.34). The resulting coupling improvements translate into a 24% and 13% decrease in RMSE and rBIAS, respectively. The enhanced precipitation forecast will directly impact land surface processes. The degree of improvement achieved here is in line with the findings of Givati et al. (2016) over the Aaylon Basin in Israel during a 1-month simulation of WRF versus WRF/WRF-Hydro, in which they recorded a 21% decrease in the coupled precipitation RMSE. Fig. 4 shows the observed, simulated (WRF and WRF/WRF-Hydro), and GPM-retrieved cumulative rainfall during the event at four selected stations, namely, the coastal Abu Dhabi station (4a), inland Al Ain station (4b), and mountainous Jabal Hafeet (4c) and Jabal Mebreh (4d) stations situated on the northeastern highlands at altitudes of 1080 and 1433 meters (see Fig. 1), respectively. The stations were selected to capture the spatial variability of the rainfall across the study domain as well as the potential impact of the topography with stations located at flat open terrain and other at higher altitude. In Fig. 4, time intervals of 15 minutes were used to derive the cumulative plots, while 30-minute intervals were available for the GPM product.

Table 3 summarizes the error measures at each station. In all three cases, the simulated WRF rainfall recorded the poorest performance and highest deviations (underestimation and overestimation) from the observed rainfall pattern with rBIAS values of 0.84, -0.41, 1.46, and -0.79 at Abu Dhabi, Al Ain, Jabal Hafeet, and Jabal Mebreh stations, respectively. GPM recorded overestimations across all four stations with rBIAS measures of 0.74, 0.22, 0.86, and 1.53 at stations of Abu Dhabi, Al Ain, Jabal Hafeet, and Jabal Mebreh, respectively. Both models show a 4 to 6-hour latency in rainfall initiation between 18Z 08/03/16 and 23Z 08/03/16. However, after the initiation phase, the WRF-Hydro system follows the observed patterns more closely, while outperforming (in terms of RMSE and rBIAS) the GPM pattern at the stations of Abu Dhabi (0.9 and 0.11 vs. 1.1 and 0.74) and Jabal Mebreh (0.38 and -0.35 vs. 0.96 and 1.53). Despite the proximity of the Al Ain (b) and Jabal Hafeet (c) stations, the former's observed rainfall accumulations are magnified by approximately a factor of 2. This is attributed to the effect of topography with their elevation difference reaching 957 m and the location of the Jabal Hafeet rain gauge, situated on the lee side of the mountain with respect to the advection of the storm, whereas the Al Ain gauge is in an open desert terrain with no topographic obstructions.

#### 4.1.2 Station 2-meter Temperature versus WRF and WRF/WRF-Hydro

Fig. 5 shows the hourly observed and simulated WRF and WRF/WRF-Hydro 2-meter temperature at the stations of Abu Dhabi (5a), Al Ain (5b), Jabal Hafeet (5c), and Jabal Mebreh (5d). Both WRF and WRF/WRF-Hydro show a strong warm bias across all 4 stations during the morning and day hours, with the higher biases occurring during the first day of the simulation (08/03/2016). Also, both models produced smaller cold biases during the night hours. The stations recorded a sharp decrease

in temperature between 07Z 09/03/16 and 10Z 09/03/16 which is temporally consistent with the associated rainfall initiation at each station in Fig. 4. Both simulations with WRF\WRF-Hydro and WRF reproduced the decline in temperature especially at Abu Dhabi (5a) and Al Ain stations (5b). The decrease in temperature between 07Z 09/03/16 and 10Z/09/03/16 at the mountain stations, namely, Jabal Hafeet (5c) and Jabal Mebreh (5d) was not significant. Overall, the difference between WRF\WRF-Hydro and WRF temperatures was more significant at Al Ain and Abu Dhabi stations than the mountain stations. The decrease in temperature simulated by WRF\WRF-Hydro and WRF between 07Z 09/03/16 and 10Z/09/03/16 preceded the decline in the observed temperatures, which is in line with the lag between simulated and observed rainfall, especially at the Abu Dhabi station with simulated rainfall initiated earlier than the observed.

Fig. 6 shows the scatterplots of the hourly 2-meter temperature from the WRF and WRF/ WRF-Hydro output fields against the temperature observations from all 57 stations in the network. The warm biases appear to increase with the rise in temperature, confirming the previously noted stronger daytime biases and its consistency at all stations.

Table 4 lists the statistical measures obtained from both simulations compared to the observed records. Slight improvement (below 1 $^{o}$C) was achieved in the simulated range (min/max) and mean by the coupling. Similarly, minor improvements were recorded in the PCC (0.81 vs. 0.71), RMSE (1.56 versus 1.61) and rBIAS (0.03 versus 0.04). The morning overestimation of 2-meter temperature can be attributed to the dual cooling effects of existing dust/aerosols and the strong land-sea breeze interactions noted by Lazzarini et al. (2014). Both factors are dominant over coastal and arid regions, such as the UAE, and are not fully incorporated in the current model physics.

The increase in rainfall should have increased soil moisture and therefore led to an increase of the latent heat over the sensible heat, causing a decline in air temperature. The improvement of temperature simulation with WRF-Hydro over WRF could be attributed to the capability of WRF-Hydro to simulate soil moisture spatial distribution more reliably than WRF. Nevertheless, both models still show a warm bias. The heat exchange between land and atmosphere which controls the change in air temperature is site specific. It also depends on land cover conditions which defines the roughness length for heat and local topography (macro roughness and surface geometry) which defines the roughness length for momentum (Yang et al., 2008). This is also in combination with the soil moisture effect which impacts surface emissivity, surface temperature, and therefore, the simulated air temperature. This explains the different behaviors of temperature at the different sites.

#### 4.1.3 Station Global Radiation versus WRF and WRF/WRF-Hydro

Fig. 7 shows the hourly observed and simulated WRF and WRF/WRF-Hydro global radiation at the stations of Abu Dhabi (7a), Al Ain (7b), Jabal Hafeet (7c), and Jabal Mebreh (7d). Both WRF and WRF/WRF-Hydro show overestimations, especially for the Jabal Mebreh station, over the first day of the simulation (08/03/2016). Similar to the 2-meter temperature warm bias evolution, the radiation overestimations are reduced in the second day of the simulation (09/03/2016). Also, the lower radiation readings during the second day are temporally consistent with the 2-meter temperature depressions (Fig. 5) and rainfall initiation (Fig. 4) at each station. By inspection, the WRF/WRF-Hydro simulation better matches the observed temporal variation and magnitudes than the WRF simulation.

Fig. 8 shows the scatterplots of the global radiation from the WRF and WRF/WRF-Hydro simulations fields against the station observations. The WRF/WRF-Hydro radiation variability demonstrates much less deviation from the station observations compared to that of the WRF model.

Table 5 lists the statistical measures obtained from both simulations compared to the observed records. Higher agreement was achieved by the coupling in the mean and maximum of the simulated radiation with a mean of 112.68 versus 133.31 W/m$^2$ compared to the 101.96 W/m$^2$ mean of observations and a maximum of 805.12 versus 788.16 W/m$^2$ compared to the 985.05 W/m$^2$ maximum of observations. The coupling improvement is further corroborated by the PCC of 0.89 versus 0.78, RSME of 73.72 versus 139.61, and rBIAS of 0.21 versus 0.33. This is potentially explained by the lower day-night amplitude in surface temperature with WRF-Hydro, which, in turn, reduces the deviation in upward longwave radiation and net radiation.

Zempila et al. (2016) assessed the performance of different shortwave radiation schemes, namely, the Dudhia, updated Goddard and the Goddard Fluid Dynamics Laboratory (GFDL), and RRTMG (used here). They compared the simulated global radiation to a set of hourly measurements at 12 stations over Greece. Overestimations between 40 to 70% were recorded for all schemes, while better agreement was achieved during clear (cloudless) sky conditions. In the present study, improved bias results of 33% and 21% biases were obtained from WRF and WRF/WRF-Hydro, respectively (see Table 5).

Fig. 9 shows the cumulative cloud fraction from the MODIS level 2 retrievals (10-km) and both the WRF and WRF/WRF-Hydro simulations (1-km) for March 9, 2016. The cloud fraction ranges from 0 (cloud-free) to 1 (complete cloud cover). The cloud base altitude was found to be in the range of 5 km from both MODIS, standalone WRF, and WRF/WRF-Hydro, which is also in line with the profiles of Fig. 2. Overall, the simulated cloud cover reproduced much smaller extents, especially, overland in the western coastal areas. Díaz et al. (2015) assessed the capability of the WRF model to reproduce clouds over the African region with varying configurations and physics options and their comparisons with satellite observations. They concluded an overall underestimation of cloud cover from their 9 WRF simulations, particularly, in the case of marine-boundary layer clouds over coastal areas. Their simulations resolved a high number of thick clouds and too few clouds with lower optical thickness. The net result was an underestimation of low cloud cover, which is the case of the present study. Otkin and Greenwald (2008) also examined the ability of WRF to reproduce cloud properties during an extratropical cyclone over the North Atlantic Ocean. Similar to the present study, they relied on MODIS retrievals for model verification with different combinations of cloud microphysics and PBL schemes, and found consistent underestimation. They attributed the underestimation to the utilization of radiance and reflectance data on a 1-km grid by MODIS and, therefore, its ability to capture small cumulus clouds, whereas the WRF model horizontal resolution (4 km) failed to explicitly resolve all processes. This is not the case with the present study, given the matching 1-km horizontal resolution used for d03. Hence, model resolution is shown to be of less significance for cloud resolving, while focus should be placed on the inclusion of a cumulus parameterization (not explored here) which may improve the model simulations through a better representation of the subgrid-scale cumulus clouds within this region. It is also possible that MODIS overestimated cloud coverage over the study domain. It is known that in the presence of high reflective surfaces (high albedo) in the background like snow/ice or desert cloud products become less accurate (Kotarba, 2010).

As a key factor in land-atmosphere interactions, clouds directly impact the radiation balance in terms of the amount reflected, absorbed, and emitted, depending on various cloud physical properties. Therefore, the underestimation of cloud fraction in both model configurations explains the observed overestimation in simulated global radiation, with less reflectance and more radiation reaching the surface. Also, the underestimation of precipitation during the first 12 hours (see Fig. 4) is primarily attributed to the models' spin-up time and GFS bias during initialization. This could be supported with Fig. 9 which shows an underestimation of cloud extent in the cases of both WRF and WRF/WRF-Hydro.

## 4.2 Analyses of Hydrological Processes

### 4.2.1 WRF-Hydro Soil Moisture versus AMSR2 retrievals

Fig. 10 shows the soil moisture retrievals from ASMR2 (10 km), the standalone WRF (1D Noah-MP - 1 km), and the simulated WRF/WRF-Hydro (100 m) during (2016-03-09-00:00:00) and after (2016-03-10-00:45:00) the event. The increase in soil moistures from both model simulations along the coast and western desert areas after the event (2016-03-10-00:45:00) verifies the anticipated soil exfiltration and runoff drainage direction from the northeastern mountains toward the western lowlands (see Fig. 1). The higher soil moisture values resulting from the WRF/WRF-Hydro simulation are primarily attributed to the increase in simulated precipitation compared to standalone WRF (see Fig. 3).

Fig. 11 shows scatter plots of the soil moisture values from ASMR2 and WRF/WRF-Hydro from both timings after re-gridding at 10 km through least-square interpolation. The spatial comparison of WRF/WRF-Hydro and AMSR2 soil moisture estimates revealed three increasing soil moisture classes: R1, R2, and R3 (delineated in Fig. 10) with an overall RMSE and rBIAS of 0.07 and 0.08 (8%), respectively.

The region over the western part of the country (R1) received negligible rainfall (see Fig. 3), and, consequently, recorded the lowest soil moisture value class ranging from 0 to 0.15 $m^3/m^3$. Hence, the default USGS soil conditions and parametrization in the Noah-MP scheme remained unchanged, while showing the highest positive overestimation with an rBIAS of 0.64. The second class of soil moisture values (R2), ranging from 0.15 to 0.25 $m^3/m^3$, received light simulated rainfall between 30 to 70 mm (see Fig. 3). This class recorded the lowest positive rBIAS of 0.02, which is attributed to the negating effect of the simulated rainfall underestimations on the existing positive initialization biases. Whereas, the third class of soil moistures (R3), ranging from 0.25 to 0.5 $m^3/m^3$, recorded a negative bias of -0.18. This can be explained by the combined effect of higher simulated rainfall underestimations and the topographical corrections incorporated in the AMSR2 product, resulting in outliers beyond 0.35 $m^3/ m^3$. The uncertainty of passive microwave retrievals over rough terrain has been recorded by several studies (Park et al., 2016;Zhan et al., 2015;Wang et al., 2010;Njoku and Chan, 2006). Moreover, the use of passive microwave C and X bands frequencies in the retrieval of soil moisture should only reflect the effect of water content in the top 1 cm of soil as the penetration of the signal is limited, especially in the case of wet soils. This shallow measurement fails to match with the 10 cm depth of the first layer (10 cm) in WRF-Hydro (Ek et al., 2003). The increase of soil moisture in R2 and R3 implies that WRF-Hydro simulated the routing of the streamflow in wadis and the lateral flow of subsurface saturated soil. The increase

could also be attributed to exfiltration from saturated soils of water flowing from the Hajar Mountain region towards lower lands in the western region. Recall that simulations were carried out in arid regions with ephemeral and ungauged rivers. In the absence of streamflow and/or water level measurements, the verification of the hydrological processes in this study relied on remotely sensed observations. We mainly relied on passive microwave retrievals of soil moisture which are known to be more reliable and make use of well-established algorithms compared to other retrievals from active microwave or thermal satellite observations. Nevertheless, retrieval from passive microwave observation are relatively coarse in terms of spatial resolution. They remain however relevant for regional assessments like the one conducted in this study.

#### 4.2.2 WRF-Hydro Soil Moisture Propagation with Lateral Flow

The Soil Moisture Operational Products System (SMOPS), provided by the National Oceanic and Atmospheric Administration (NOAA), merges soil moisture retrievals from multi-satellites/sensors to generate a global product at higher spatial and temporal coverage (Liu et al., 2016). Relevant to the current study period, SMOPS now incorporates near-real time SMAP data and includes soil moisture retrievals from the GPM Microwave Imager (GMI). The 6-hourly product mapped at 0.25° spatial resolution is used here to assess the accuracy of the simulated soil moisture.

A comparison of soil moisture evolution at the upstream and downstream of a wadi within the study domain is expected to verify whether soil moisture transport occurs over the storm timescale. A wadi within the coverage of the Saih Al Salem station (24 49 39 N, 55 18 43 E) was selected to conduct this test. Fig. 12 shows the time series of simulated soil moisture from WRF/WRF-Hydro at two locations upstream and downstream of the wadi. SMOPS retrievals are overlaid as data points, along with the hyetograph recorded at the corresponding Saih Al Salem station at the top. Given the short distance (less than 1km) separating the two locations, a lag time of less than 1 hour is observed between the two soil moisture peaks. The first rain of approximately 22 mm at 22 Z 08/03/16 triggers an immediate increase in soil moisture from 0.18 to 0.25 $m^3/m^3$. The subsequent rainfall then elevates the moisture further to around 0.34 $m^3/m^3$, with a slight increase in the peak of downstream soil moisture compared to that of the upstream that could be attributed to additional lateral drainage. However, at 18Z 09/03/16 the downstream soil moisture rises again to a sustained peak at around 0.32 $m^3/m^3$, while the upstream soil moisture continues to dissipate through infiltration and evaporation. In the absence of additional rainfall, this sustained peak in downstream soil moisture is the result of lateral surface flow from the upstream which is resolved by WRF-Hydro and fed back to the soil moisture fields. Despite the SMOPS data gaps during the event, the merged retrievals consistently increase during the event with reasonable accuracy compared to the simulated soil moisture fields. SMOPS data have a coarse spatial resolution that is an inherent limitation related to the resolution of the passive microwave signal, which does not allow verifying the drainage between the selected points. Nevertheless, the product captured the increase of moisture that is a result of the event and the persistent plateau of soil moisture after the event which seem to be in a better agreement with WRF-Hydro values.

#### 4.2.3 Soil Moisture – Precipitation Feedback

An increase in water content of the top soil layer decreases both the surface albedo and the Bowen ratio. A lower surface albedo dictates more absorbance of net radiation, while lower Bowen ratios are a result of higher water vapor content in the boundary layer and more downwards flux of terrestrial radiation at the surface due to the water vapor greenhouse effect. This dual effect amounts to a larger total flux of heat from the surface into the boundary layer (Eltahir, 1998). Furthermore, the cooling of surface temperature accompanied by the moisture should be associated with a reduced sensible heat flux and a smaller PBL height. Fig. 13 shows the PBL heights from both simulations with larger collapses resolved from the coupled model. According to Seidel et al. (2010), PBL heights can be inferred from radiosonde data (Fig.2 ), particularly based on determining maximum or minimum vertical gradients of relative humidity or specific humidity. Such methods yield better agreement compared to those relying on locations of elevated temperature inversions or mixing height. Hence, using the Abu Dhabi radiosonde profile of Fig. 2, and based on the gradient approach, the PBL height can be estimated to be in the range of 90 – 200 at 12 Z, which is closer to that simulated from the coupled WRF/WRF-Hydro (190 m) compared to standalone WRF (750 m).

Similar to the present study, Xiang et al. (2017) used the coupled WRF-Hydro system for short-term (72-hour) simulations of storm events to discretized the effects of higher soil moisture conditions on precipitation generation, using the framework proposed by Eltahir (1998) to diagnose mechanisms of positive soil moisture-precipitation feedback. They captured an increase of up to 26 mm in WRF/WRF-Hydro precipitation over 48 hours, which is in line with the present study with a maximum increase of 23 mm.

Koster et al. (2004) identified regional hot spots, including the Arabian Peninsula, and particularly the UAE, where a global initialization of soil moisture may enhance precipitation prediction skill during Northern Hemisphere summer. Assuming predominantly local soil moisture impacts, the hot spots indicate where regular monitoring of soil moisture using in situ and satellite observation may lead to an enhancement in boreal summer seasonal forecasting. Their study also referred to a main challenge related to the dependency of the models on soil moisture computational estimates, especially that a long spin up period might be required to reproduce reliable soil moisture values for seasonal forecasting. Senatore et al. (2015) showed that simulations for one month in a watershed in Italy required a two month spin up period. This fosters the importance of deploying dense soil moisture monitoring networks in the region (AlJassar et al., 2015;Temimi et al., 2014;Fares et al., 2013) which should contribute to a better understanding and characterization of soil moisture variability and hydrological processes in desert and hyperarid environments. The description of routing and lateral flow by WRF-Hydro improved the quality of the simulated atmospheric processes in this case study. This promises improved seasonal precipitation forecasts, as well as short-term predictions assessed here.

## 5 Conclusions

In this study, we simulated an extreme weather event in March, 2016 over the UAE, a country within the Arabian Peninsula of particular interest for hydrometeorological research and monitoring. The event was simulated from both standalone WRF and fully coupled WRF/WRF-Hydro model configurations and compared to station observations and ongoing satellite products. The main objective of the study was to investigate the added value of coupled land surface-atmospheric modeling for precipitation forecasts over the hyper-arid environment of the UAE, while employing current modeling tools to aid in operational forecasting efforts in the region.

Results showed reductions of 24% and 13% in RMSE and rBIAS measures, respectively, for precipitation forecasts from the coupled model configuration. Furthermore, the coupled WRF/WRF-Hydro system was found to outperform GPM rainfall retrievals at some stations (e.g. Abu Dhabi and Jabal Mebreh). The demonstrated improvement in coupled precipitation simulation, at the local scale, greatly enhances the accuracy of hydrologic forecasts and flash flood guidance systems. Senatore et al. (2015) explained the higher precipitation with their WRF/WRF-Hydro simulation by the differences in surface temperatures as warmer surface boundary condition may lead to more convection and therefore higher energy and rainfall. The same interpretation could be also adopted in this study. Nevertheless, the deep intrinsic factors, primarily the impact of internal atmospheric variability (Rasmussen et al., 2012), causing this improvement remain subjects of current research and were not explored here (Givati et al., 2016;Senatore et al., 2015). The lateral boundary conditions are expected to severely restrict our model during this short 48-hour simulations, however internal model variability is a direct consequence of the non-linear dynamical and physical internal processes being active and detectable (Christensen et al., 2001). However, despite the more skillful forecasts of the coupled system, the bias remains high (21%), which dictates the need for ongoing hydrometeorological forecast enhancement.

The coupled system also showed improvements in global radiation forecasts (45% and 12% for RMSE and rBIAS, respectively), while less significant enhancements were observed in the case of surface temperature (3.1% and 1%). Both parameters were subject to high positive biases during the morning and daytime. The warm temperature biases were attributed to dry biases in the NCEP-GFS boundary conditions observed by Chaouch et al. (2017) and Yang et al. (2011), and the uncaptured cooling mechanisms of aerosols and sea breezes, while the underestimation of cloud cover explained the overestimations in global radiation. The diurnal temperature signal was not captured, even in the observed values, due to the extreme event. Also, the discrepancies in temperature simulation could be caused by soil moisture simulation and its spatial organization within the study domain which impacts the latent heat, the heat exchange, and therefore the temperature difference. More importantly, higher spin up times (6 hours used here) can add to the model accuracy in terms of both atmospheric dynamics and hydrological processes (Lo et al., 2008). Soil moisture validation – a challenging application over arid regions – showed varying response classes across the UAE, and were consistent with the expected surface flow directions. The difference of spatial scales between the retrieved and simulated soil moisture, and the impact of high reflectance from

desert land cover on the AMSR2 microwave retrieval algorithms may have contributed to the observed discrepancies (Wehbe et al., 2018).

The fully coupled model configuration captures the complete dynamics of the water and energy cycles, starting from the upper atmosphere to the unsaturated and saturated zones on the land surface, and back. Land surface-atmospheric interactions are primarily governed by two key hydrological parameters, namely rainfall and soil moisture. Hence, future work with in-situ soil moisture data assimilation is expected to enhance the model accuracy, both overland and in the atmosphere, through the captured feedbacks. This case study focused on a regional event triggered by a large scale system. Hence, the impact of accounting for hydrological processes through the online coupling is not expected to be significant. Nevertheless, an improvement in the simulation of precipitation was obtained with the coupled WRF/WRF-Hydro model. To further discretize the added value of the coupling demonstrated in this study, ensemble approaches should be the focus of future work in order to assess the robustness of the potential improvements with WRF-Hydro. In the present study, the exact contribution of lateral flow versus internal atmospheric variability on the captured improvement remains an open question that is subject to further research in this area, particularly for arid environments that have not been receiving much attention.

*Data availability*. The GPM data is provided by the NASA/Goddard Space Flight Center and archived at the NASA GES DISC (www.pmm.nasa.gov). The AMSR2 data is produced by Remote Sensing Systems and sponsored by the NASA AMSR-E Science Team and the NASA Earth Science MEaSUREs Program (www.podaac.jpl.nasa.gov). The NCEP GFS Research Data is archived at the National Center for Atmospheric Research, Computational and Information Systems Laboratory, Boulder, CO (www.rda.ucar.edu). The MODATM product is retrieved from the online data pool, courtesy of the NASA EOSDIS Land Processes Distributed Active Archive Center (LP DAAC), USGS/Earth Resources Observation and Science (EROS) Center, Sioux Falls, South Dakota (www.lpdaac.usgs.gov). The ground-based observations are obtained from the UAE National Center of Meteorology (NCM), which were provided under an agreement with clauses for non-disclosure of data. Access to this data is restricted and readers must request it through contacting research@ncms.ae.

*Author Contributions*. MT and YW conceived and designed the case study. YW performed the simulations, post-processing and data analyses. YW and MT analysed the results and wrote the article. MW, NC, OB, and TS advised on compiling and configuring the model in coupled mode. VW reviewed the results and revised the article. XZ and JL provided the NOAA SMOPS data and reviewed results. AA ensured quality control of the weather station observations.

*Competing Interests*. The authors declare that they have no conflict of interest.

*Acknowledgements*. The authors acknowledge Khalifa University of Science and Technology (KUST) for the provided support. The authors would also like to thank the UAE National Center of Meteorology (NCM) for the quality-controlled weather station observations used in this study. This work was conducted within the framework of a project funded by the NCM under the UAE Research Program for Rain Enhancement Science (UAEREP). The authors thank the NCM Executive Director, Dr. Abdulla Al Mandous, and the UAEREP Program Director, Alya Al Mazroui, for their continued support.

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

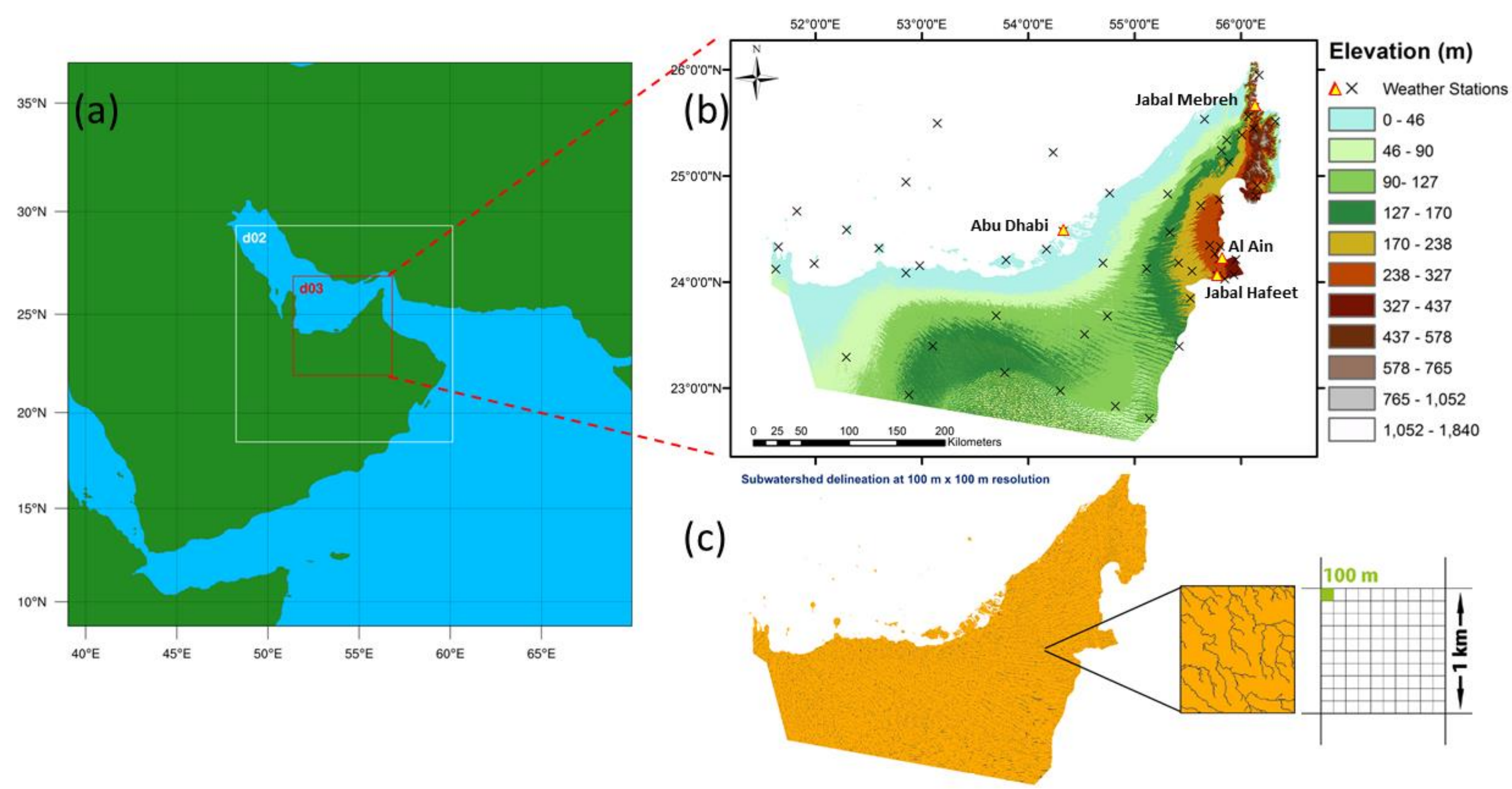

**Figure 1.** WRF model nested domains at 9 km, 3 km (d02) and 1 km (d03) horizontal resolutions (a), 30-meter ASTER DEM with station locations (b), and 100-meter WRF-Hydro grid derived over the UAE.

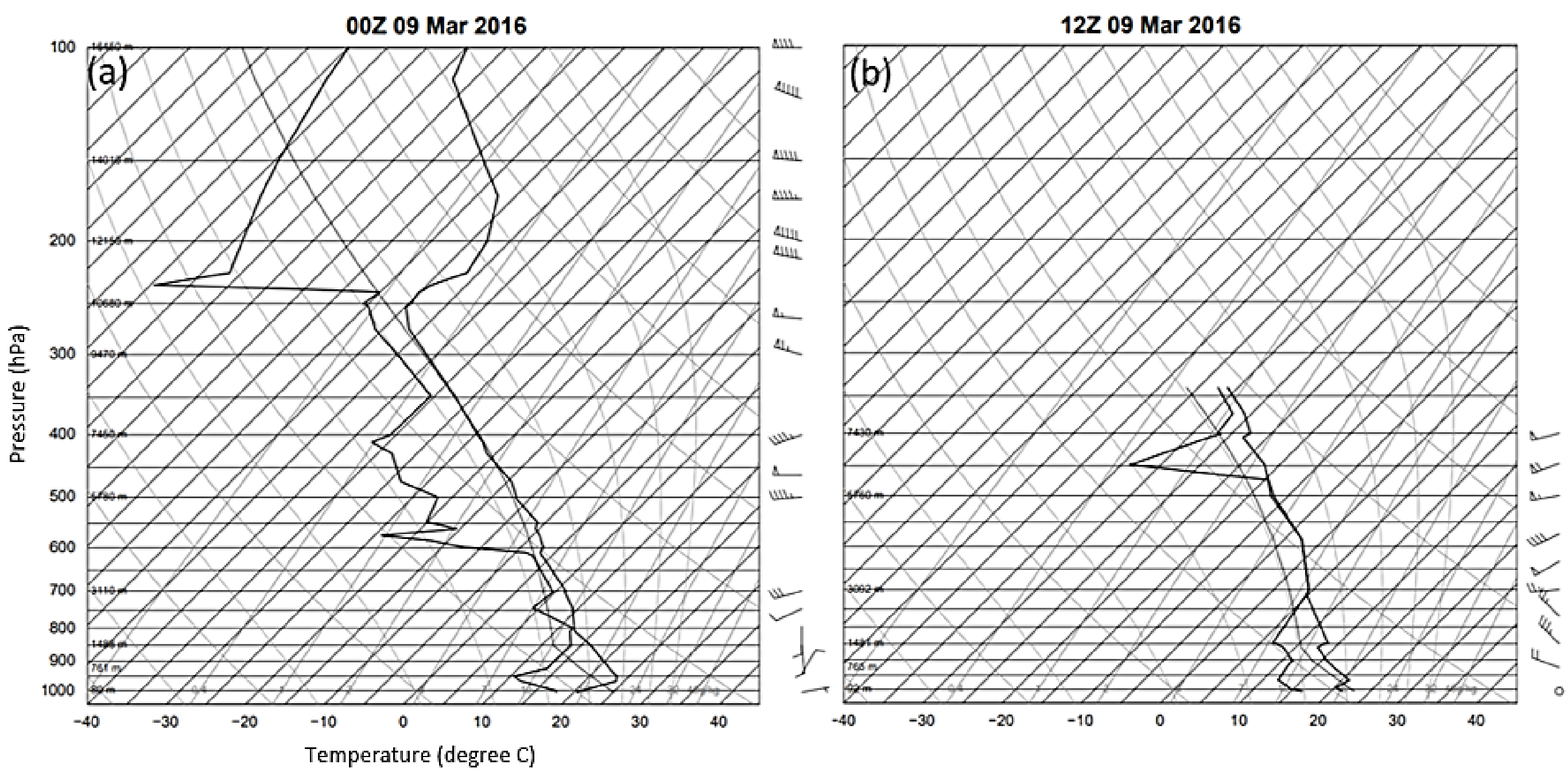

**Figure 2.** Radiosonde skew-T profiles retrieved at Abu Dhabi Airport on March 9, 2016 at 00Z (a) and 12Z (b).

**Table 1.** WRF and WRF-Hydro model configurations.

| | |
|---|---|
| **WRF and WPS version** | Version 3.7.1 released on August 14, 2015 with latest bug fixes |
| **WRF-Hydro version** | Version 3.0 released on June 14, 2015 |
| **Domain horizontal resolutions** | 9 km for d01<br>3 km for d02<br>1 km for d01, and 100 m for Hydro domain |
| **Domains horizontal grid dimensions** | 350 × 350 for d01<br>403 × 403 for d02<br>562 × 562 for d03 |
| **Projection** | Transverse Mercator |
| **Number of vertical levels** | 50 for each domain |
| **Top Pressure value** | 20 hPa |
| **Lateral boundary conditions** | GFS 6-hourly forecasts at 0.5° |
| **Initial conditions** | 6-hour spin up |
| **Longwave and Shortwave Radiation** | RRTMG |
| **Surface Layer** | Monin-Obukhov |
| **Land surface model** | Noah-MP |
| **Planetary boundary layer scheme** | Yonsei University |
| **Microphysics** | Morrison double-moment |

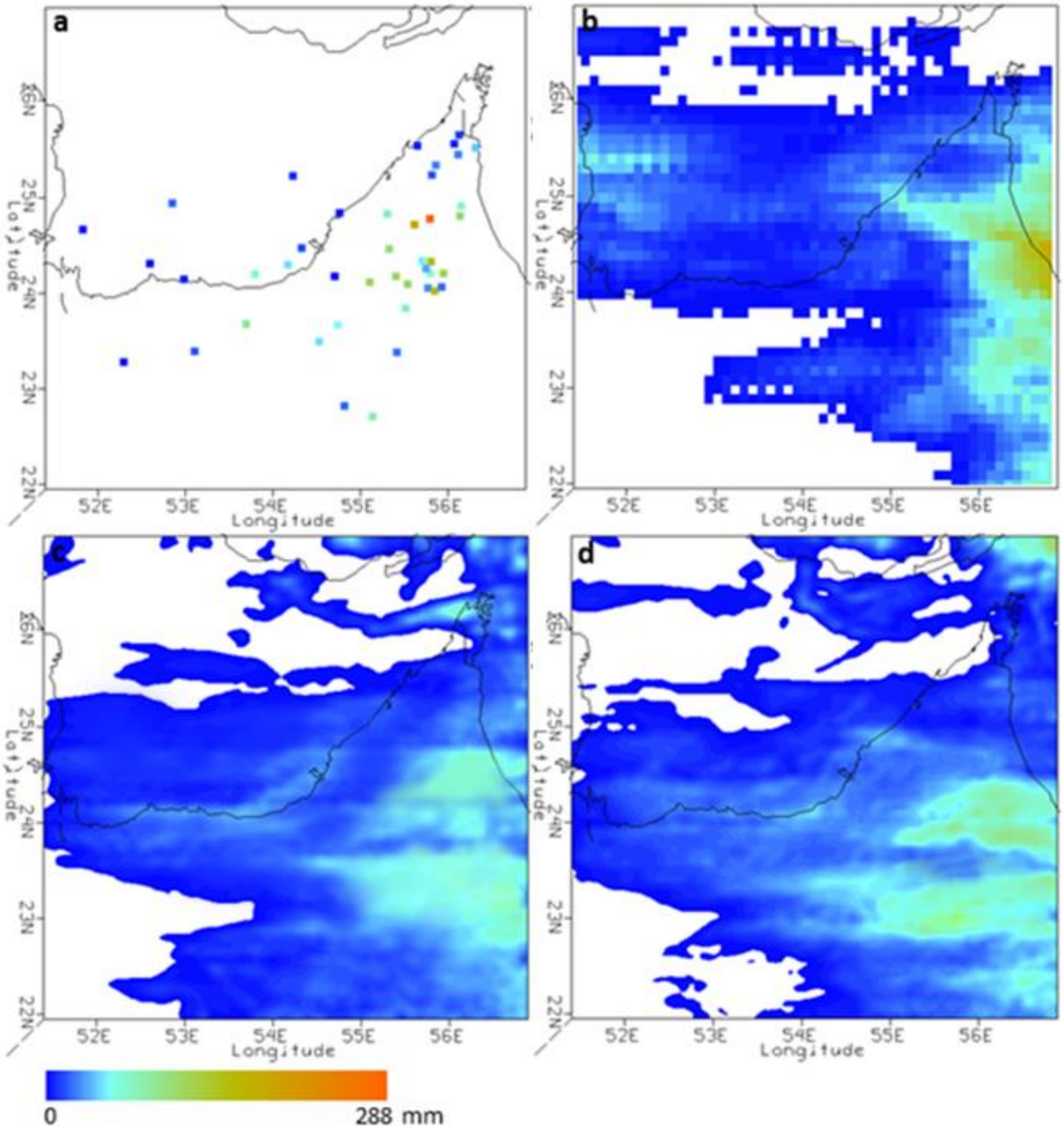

**Figure 3.** Accumulated storm rainfall over the UAE (d03) from station observations (a), GPM 30-min, 0.1° retrievals (b), WRF (c), and WRF/WRF-Hydro (d).

**Table 2.** Total storm rainfall statistical measures from collocated GPM retrievals, station observations, and simulation (WRF and WRF/WRF-Hydro).

| Precipitation Source | Mean (mm) | Min. (mm) | Max. (mm) | Std. (mm) | PCC | RMSE | rBIAS |
|---|---|---|---|---|---|---|---|
| Station Observations | 48 | 0.4 | 288 | 57 | | | |
| GPM Retrievals | 47 | 0.03 | 189 | 55 | 0.89 | 6.12 | 0.71 |
| Coupled WRF-Hydro | 39 | 0.01 | 156 | 47 | 0.82 | 10.89 | 0.21 |
| Standalone WRF | 32 | 0.01 | 122 | 42 | 0.76 | 14.24 | 0.34 |

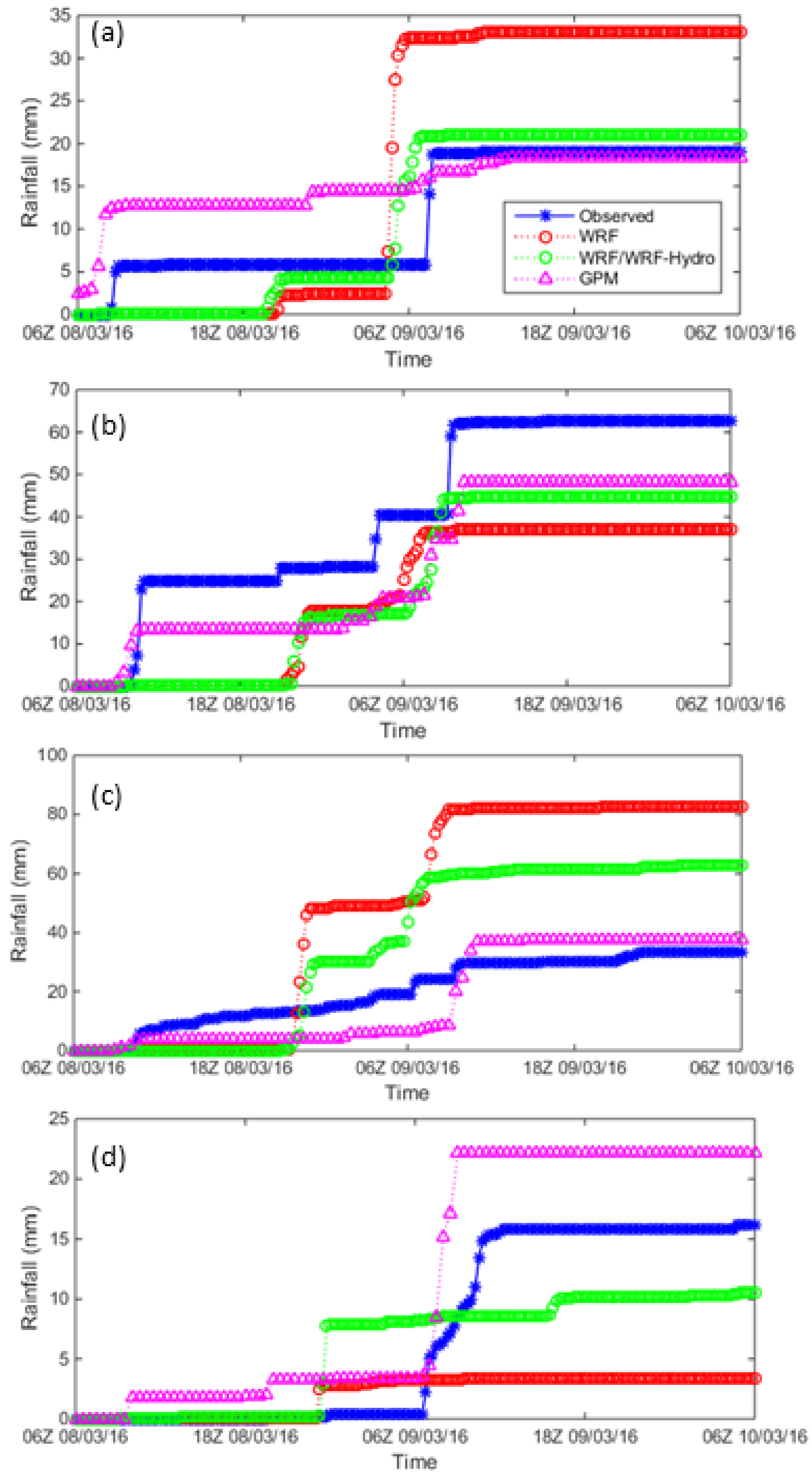

**Figure 4.** Comparison of accumulated rainfall at Abu Dhabi (a), Al Ain (b), Jabal Hafeet (c), and Jabal Mebreh (d) from stations with WRF, WRF/WRF-Hydro simulations and GPM retrievals.

**Table 3.** RMSE (rBIAS) of simulated and retrieved rainfall at Abu Dhabi, Al Ain, Jabal Hafeet, and Jabal Mebreh stations for 30-minute intervals (n = 97).

| Station | WRF | WRF/WRF-Hydro | GPM |
|---|---|---|---|
| Abu Dhabi | 1.36<br>(0.84) | 0.90<br>(0.11) | 1.10<br>(0.74) |
| Al Ain | 2.11<br>(-0.41) | 2.03<br>(-0.28) | 2.49<br>(0.22) |
| Jabal Hafeet | 1.92<br>(1.46) | 1.29<br>(0.92) | 1.46<br>(0.86) |
| Jabal Mebreh | 0.52<br>(-0.79) | 0.38<br>(-0.35) | 0.96<br>(1.53) |

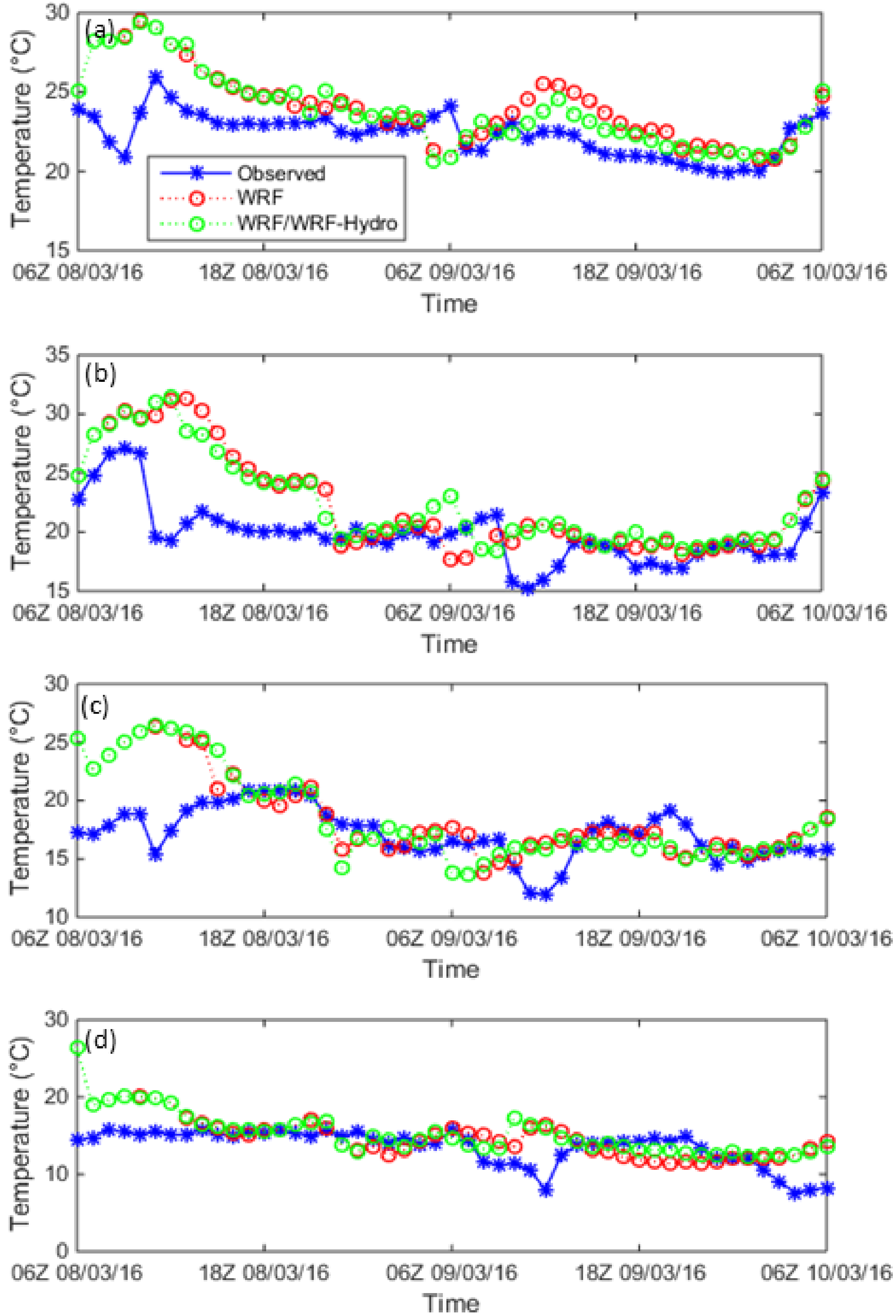

**Figure 5.** Comparison of 2-meter temperature observations at Abu Dhabi (a), Al Ain (b), Jabal Hafeet (c), and Jabal Mebreh (d) stations with WRF and WRF/WRF-Hydro simulations.

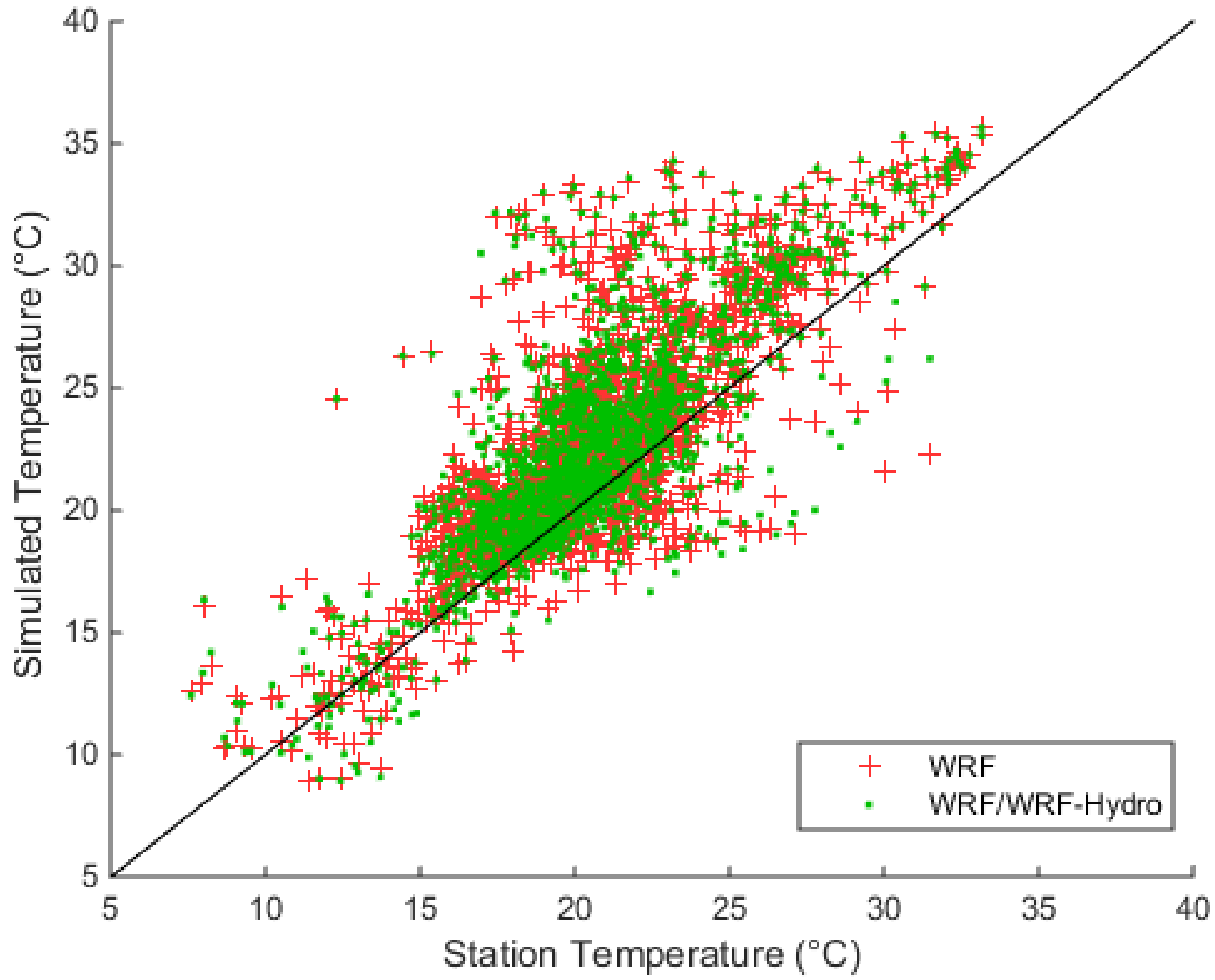

**Figure 6.** Scatterplots of 2-meter temperature from standalone WRF and coupled WRF-Hydro simulations versus station observations across the UAE.

**Table 4.** 2-meter temperature statistical measures from collocated station and simulation output.

| 2-meter Temperature Source | Mean (ºC) | Min. (ºC) | Max. (ºC) | Std. (ºC) | PCC | RMSE | rBias |
|---|---|---|---|---|---|---|---|
| Station Observations | 20.62 | 7.62 | 33.21 | 3.71 | | | |
| WRF | 22.63 | 8.91 | 35.62 | 4.64 | 0.77 | 1.61 | 0.04 |
| WRF/WRF-Hydro | 22.76 | 8.94 | 35.73 | 4.65 | 0.81 | 1.56 | 0.03 |

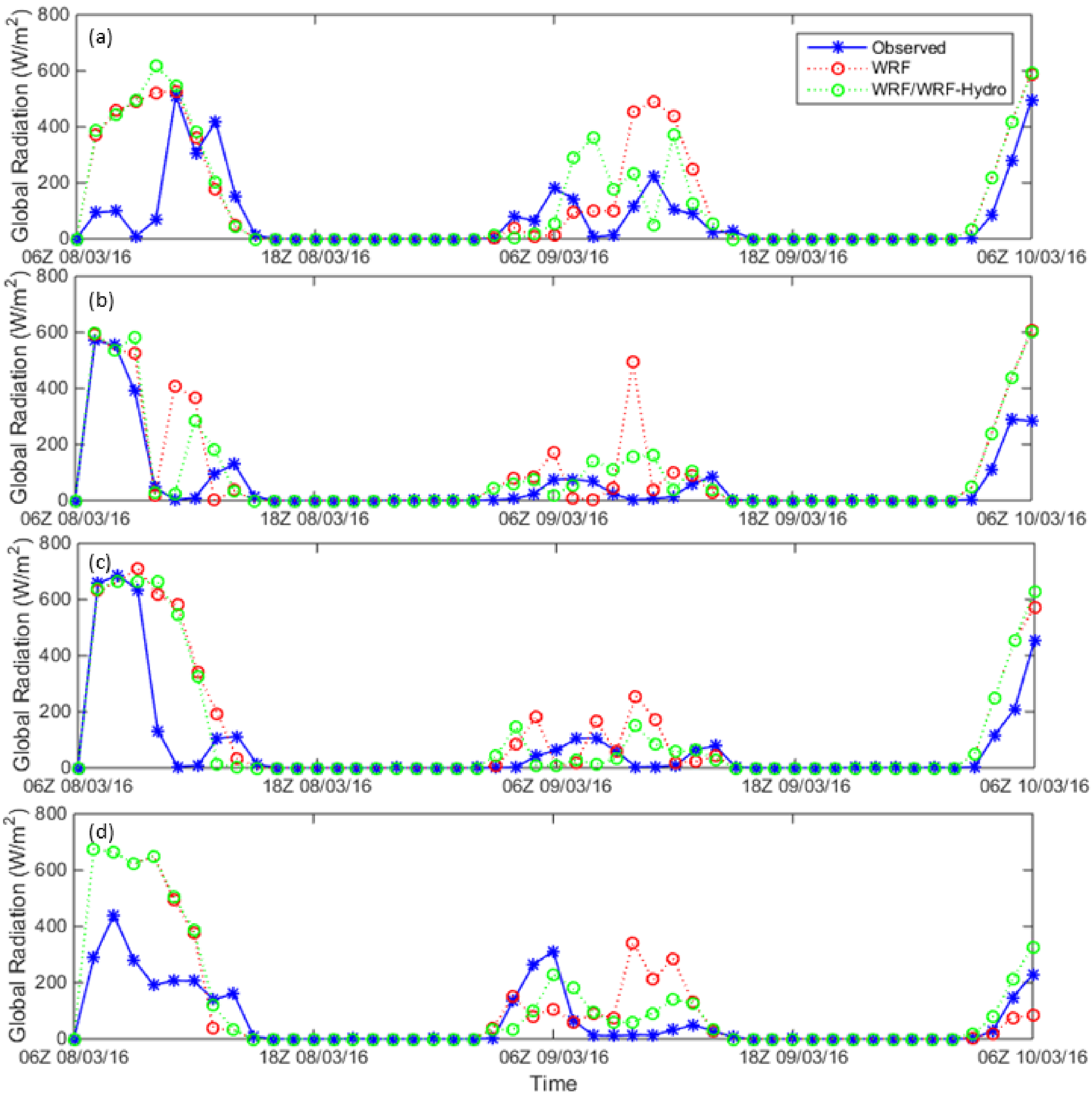

**Figure 7.** Comparison of global radiation observations at Abu Dhabi (a), Al Ain (b), Jabal Hafeet (c), and Jabal Mebreh (d) stations with WRF and WRF/WRF-Hydro simulations.

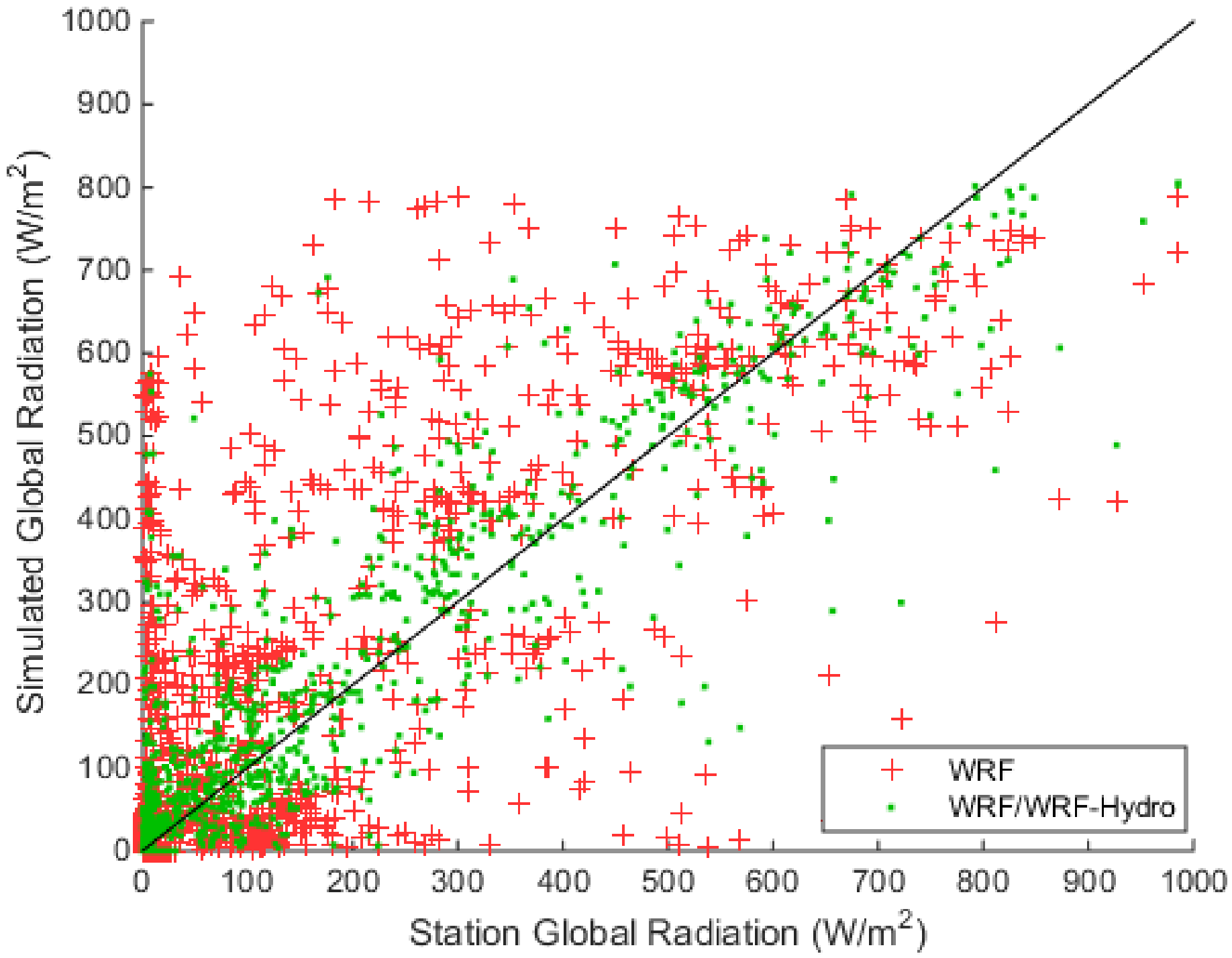

**Figure 8.** Scatterplots of global radiation from standalone WRF and coupled WRF-Hydro simulations versus station observations across the UAE.

**Table 5.** Global radiation statistical measures from the stations and simulations at 15-minute intervals (n = 194).

| Global Radiation Source | Mean ($W/m^2$) | Min. ($W/m^2$) | Max. ($W/m^2$) | Std. ($W/m^2$) | PCC | RMSE | rBIAS |
|---|---|---|---|---|---|---|---|
| Station Observations | 101.96 | 0 | 985.05 | 189.80 | | | |
| WRF | 133.31 | 0 | 788.16 | 214.70 | 0.78 | 139.61 | 0.33 |
| WRF/WRF-Hydro | 112.68 | 0 | 805.12 | 198.12 | 0.89 | 73.72 | 0.21 |

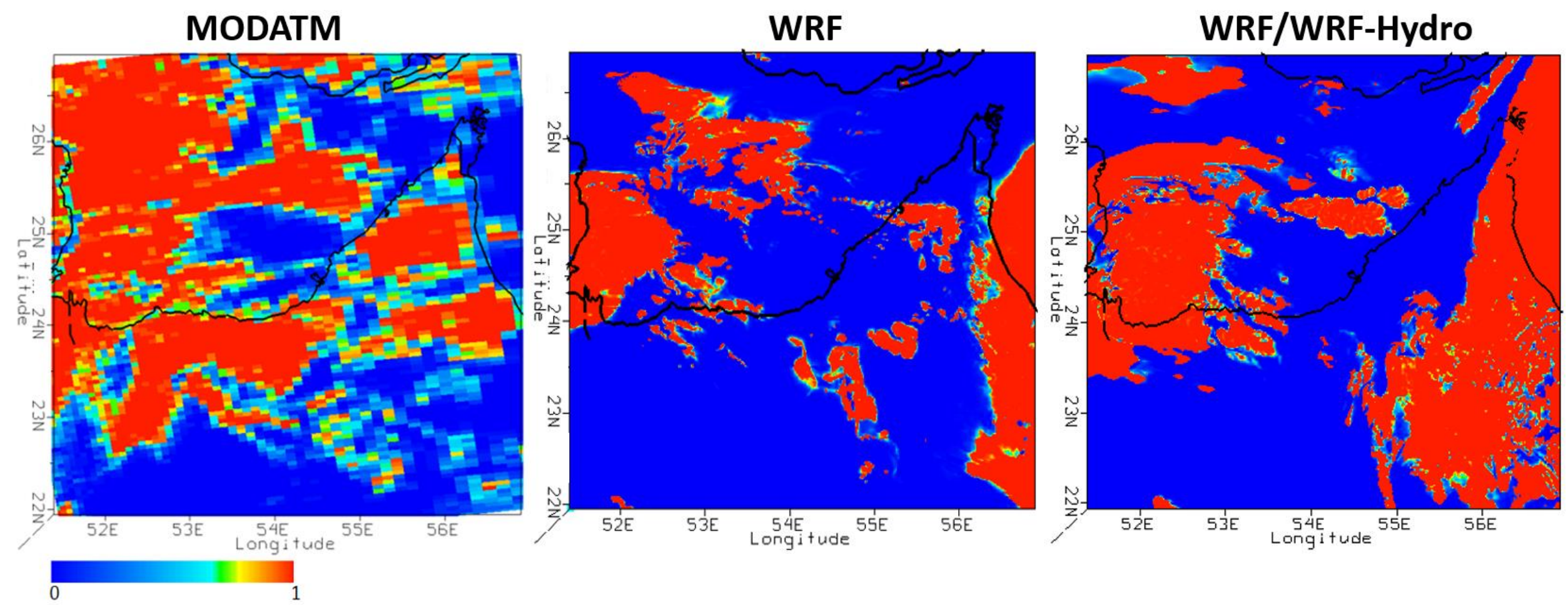

**Figure 9.** Cumulative cloud fraction retrieved from MODIS and simulated by WRF and WRF/WRF-Hydro for March 9, 2016.

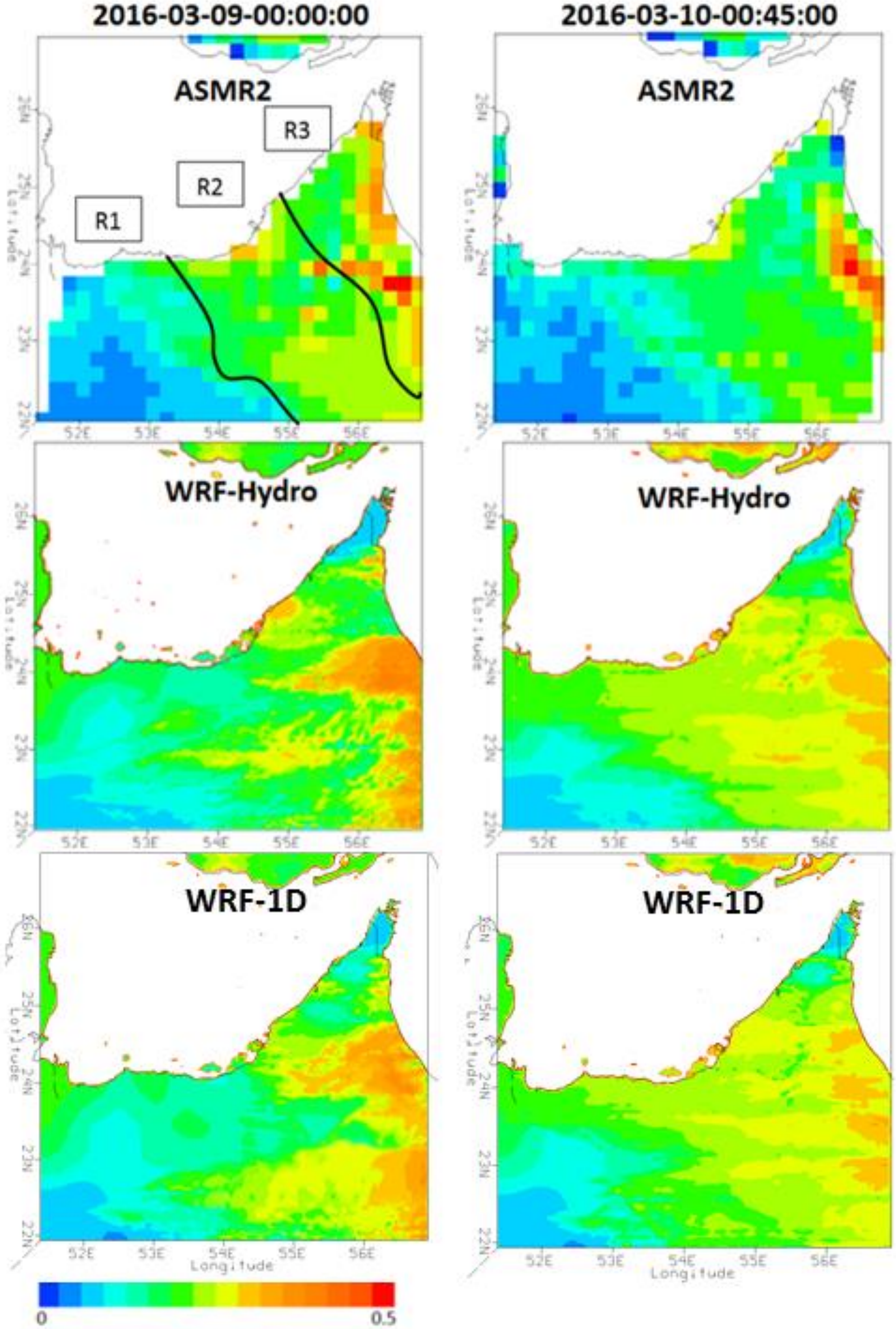

**Figure 10.** Comparison of simulated WRF-Hydro and retrieved AMSR2 soil moisture during (2016-03-09-00:00:00) and after (2015-03-10-00:45:00) the event.

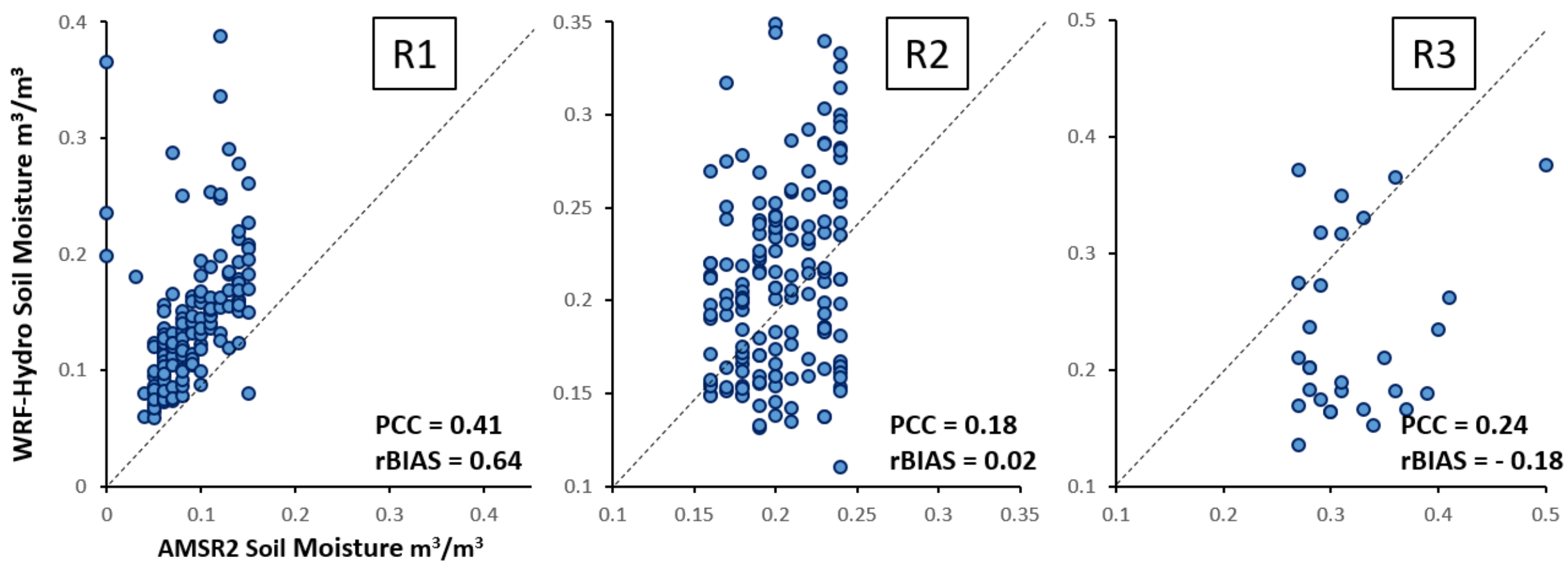

**Figure 11.** Scatterplots of AMSR2 and WRF-Hydro soil moisture estimates across the UAE.

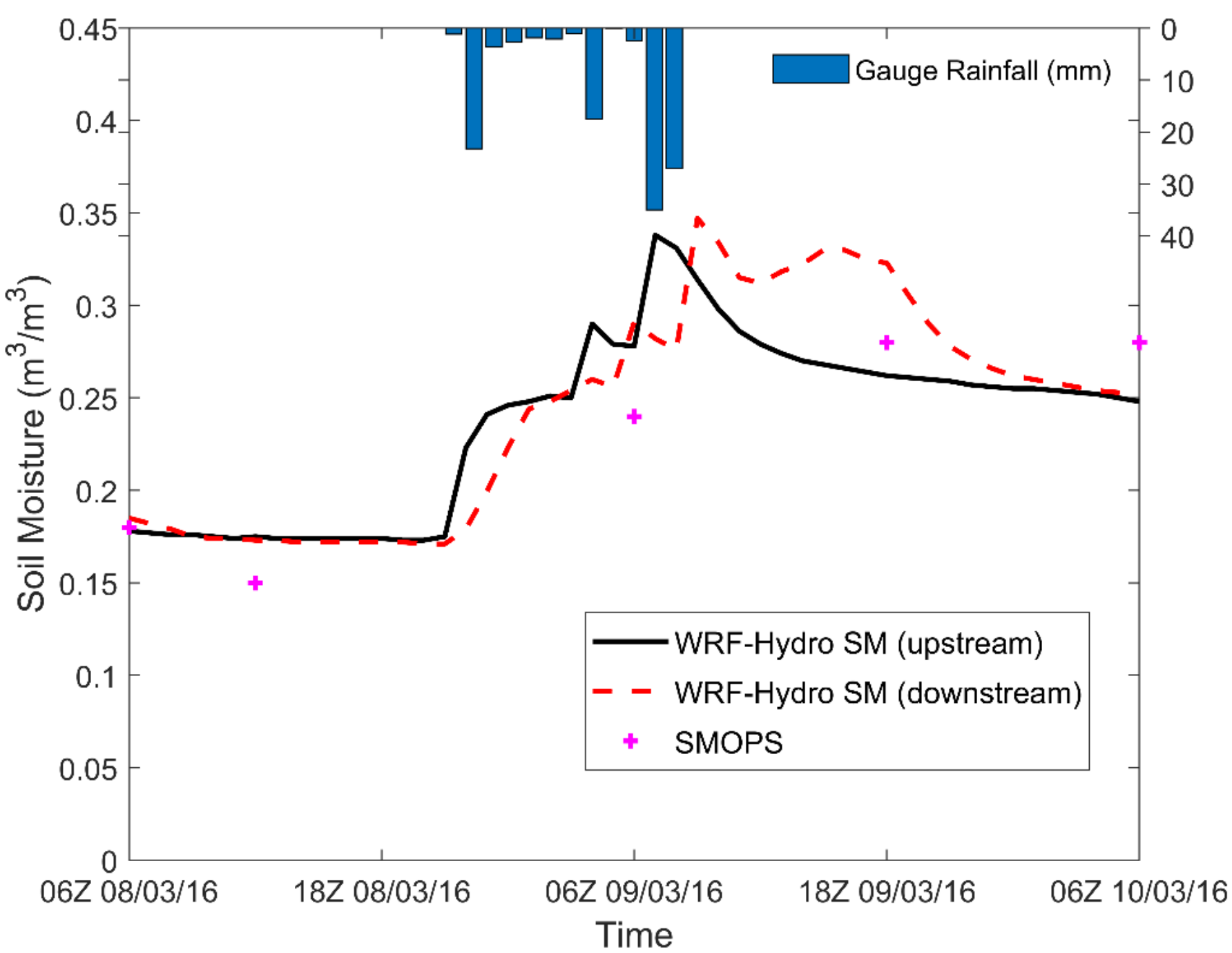

**Figure 12.** Time series of simulated soil moisture from WRF/WRF-Hydro at wadi upstream and downstream locations, along with collocated SMOPS retrievals. Hyetograph recorded at the Saih Al Salem station is shown on top.

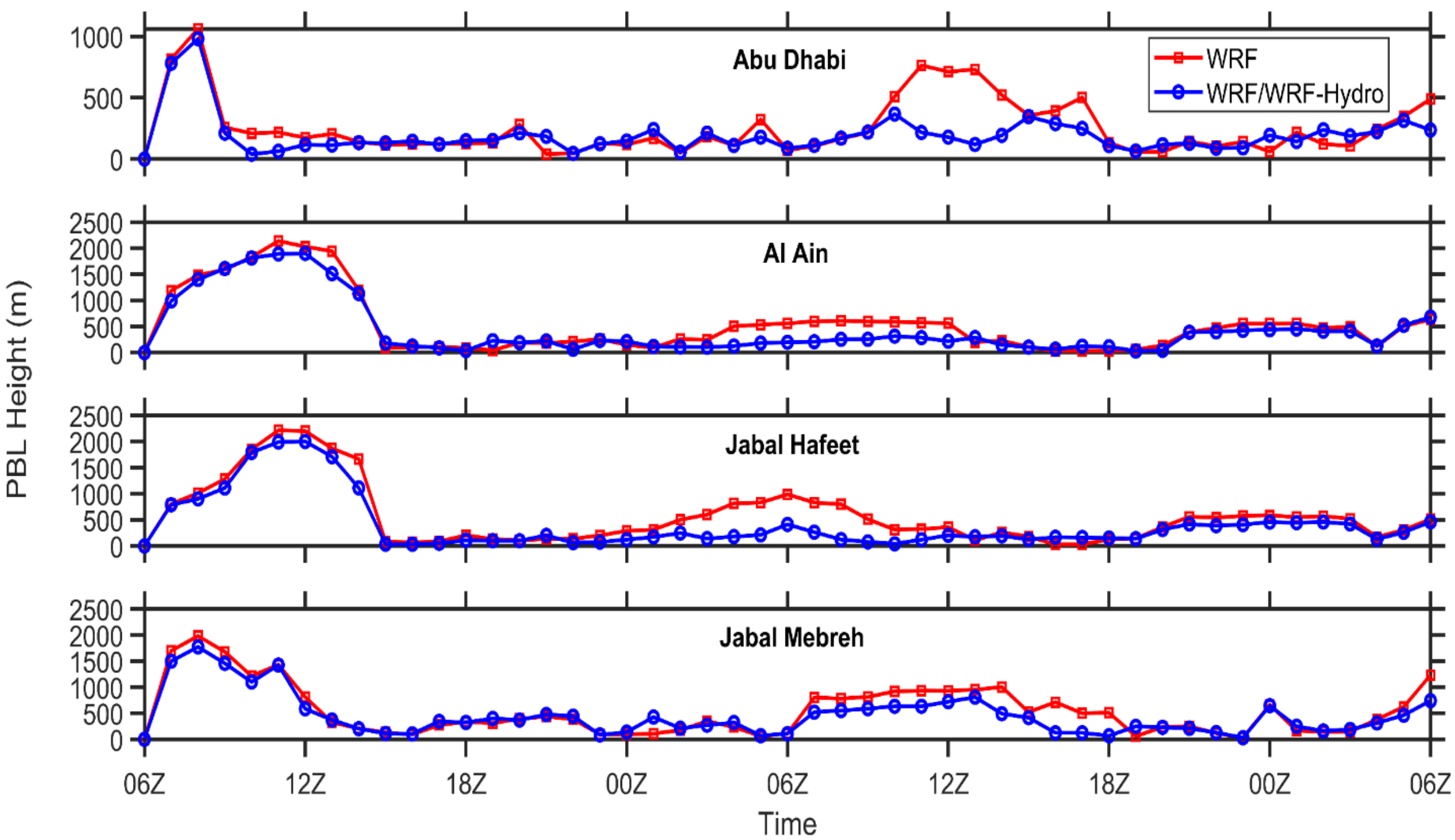

**Figure 13.** Planetary Boundary Layer (PBL) heights from WRF and WRF/WRF-Hydro simulations at each of the four stations.