# Peer review of "Analysis of an Extreme Weather Event in a Hyper-Arid Region Using WRF-Hydro Coupling, Station, and Satellite data"

_Natural Hazards and Earth System Sciences, 2018_

## Referee Comment (RC1) · N. Krakauer (Referee) · 4 Oct 2018

The manuscript presents a case study of mesoscale simulation of a severe cold-season rainstorm in eastern Arabia, where orography and small-scale variability are important factors, compared to ground-based and satellite data. I found it to be well written with a good review of relevant literature, and to make a useful contribution to a societally important concern.

My main comment is that since the standalone WRF also had the NOAH-MP land model, it is not clear to me why soil moisture feedback would improve precipitation simulations with WRF-Hydro but not standalone WRF. The difference between the two

model configurations and its relevance to the simulation performance should be discussed in more detail.

Minor and technical comments:

abstract: Use the degree sign instead of "o"

2.7: rain gauges → rain gauge

3.32 and title: hyper arid: hyphenated or one word

4.9: rephrase to "The UAE (22° to 27° N) . . . "

4.22: rephrase to "receives more rainfall compared to the country's 100 mm annual average"

5.28: clarify to what "Such frequencies" refers – L band?

7.22: "wadis" lowercase

8.30: does "analysed products" refer to ones interpolating station observations, or those from satellite retrievals or climate models?

9.10: boarder → border

13.12ff: I am not certain that WRF-Hydro includes lateral flow processes that would transport soil moisture. Please verify and give more details. Whether this transport happens in reality over the storm timescale would also need to be verified, which fits in with the authors' call for a soil moisture measurement network.

Table 2 and 4: Specify whether the GPM and model averages are collocated with the stations.

Table 3: Specify the time interval and number of data points used for the comparison between products.

2018-226, 2018.

---

## Referee Comment (RC2) · Anonymous Referee #2 · 25 Mar 2019

This manuscript presents an assessment of modelled rainfall patterns and amounts for an extreme rainfall event in UAE derived from two modelling systems, namely, the standalone WRF and the coupled WRF/Hydro system. The evaluation of model results is based on a comparison with weather stations' data (i.e. gauge rainfall data, temperature, radiation) and satellite products (i.e. the Global Precipitation Measurement (GPM) rainfall, the MODIS cloud fraction, and ASMR2 soil moisture). In the manuscript, analysed variables are limited to these hydrometeorological variables, i.e. precipitation, cloud cover, global radiation, air temperature, and soil moisture. Statistical output of the evaluation shows that the coupled WRF/Hydro is better than the standalone WRF. However, no further effort is made to diagnose the processes and mechanisms con-

trolling the water cycle that can be better captured by the coupled WRF/Hydro system than the standalone WRF. Thus I recommend that revision should be made for the following key points:

1. Literature review of the manuscript stated that numerous studies in the past have already shown the advantages of the coupled WRF/Hydro over the standalone WRF. If this study is a same kind but just a case study for another geographical location, what would be its unique contributions to knowledge?

2. As claimed in the manuscript, the main objective of the study is to investigate the added value of coupled land surface-atmospheric modeling (WRF-Hydro) over the hyper-arid environment of the UAE. In fact, the coupled WRF-Hydro system captures the dynamics of the water and energy cycles, linking the upper atmosphere to the unsaturated and saturated zones on the land surface. In order to take the full advantage of the WRF-Hydro system, diagnoses of the feedback processes/mechanisms controlling the regional scale water cycle (e.g. runoff, penetration, evaporative fraction, water vapour flux) should be conducted. Such diagnoses may lead to valuable generic outcome that could benefit the research community. In fact, the discussion in the manuscript has cited many publications for such processes/mechanisms for the purpose of interpreting the modelled output, but none of these has been further diagnosed in this study. It is strongly recommended that these diagnoses should be explored.

3. Several speculative arguments (e.g. lines 31-33 of p.10 about the processes linking rainfall to soil moisture and to 2m air temperature, lines 5-6 of p.11 about the effect of soil moisture on surface emissivity/temperature, lines 11-15 of p. 12 about resolved-scale vs subgrid scale cumulus, lines 13-15 of p.12 about underestimation of cloud by MODIS, and lines 19-20 of p. 12 about spin-up time) may be further analysed in order to show in-depth processes.

4. Figure 3 (c) & (d) and Figure 10's soil moisture plots from WRF all have shown weird
stripe structure of modelled accumulated rainfall and soil moisture, respectively. This adds doubts to model settings or post-processing and must be investigated thoroughly and the reasons should be fully explained. Once the errors are identified, all analyses should be re-done and all results should be updated.

---

## Author Response (AR1)

**Response to Reviewer 1**

We thank Reviewer 1 for the many insightful comments and suggestions. Below is our point-by-point responses to the provided comments. A marked-up version of the revised manuscript is also appended.

Reviewer Summary: The manuscript presents a case study of mesoscale simulation of a severe cold-season rainstorm in eastern Arabia, where orography and small-scale variability are important factors, compared to ground-based and satellite data. I found it to be well written with a good review of relevant literature, and to make a useful contribution to a societally important concern.

**Comment 1 (C1)**: My main comment is that since the standalone WRF also had the NOAH-MP land model, it is not clear to me why soil moisture feedback would improve precipitation simulations with WRF-Hydro but not standalone WRF. The difference between the two model configurations and its relevance to the simulation performance should be discussed in more detail.

**Author's response to C1:** We agree with the Reviewer on the need to better highlight the difference between the two model configurations and its relevance to the simulation performance.

The existing Noah-MP land surface model in standalone WRF considers single vertical columns (one-dimensional) of terrain properties at each overland grid cell (Niu et al. 2011, Ek et al. 2003). It fails to account for horizontal interactions between adjacent grid cells to calculate soil moisture, temperature profiles, runoff, and water and energy fluxes at the land surface. Therefore, the runoff–infiltration partitioning in the standalone WRF simulation is described as a purely isolated vertical process with no intake from (or discharge to) neighboring grid cells, as dictated by topography. On the other hand, the WRF-Hydro model utilizes the Noah-MP land surface model 1-D representations and attempts to improve the simulation of terrestrial hydrologic processes at high spatial and temporal resolutions by including lateral redistribution of overland and saturated subsurface flows for runoff prediction (Gochis et al. 2013) – see Fig. 1. More importantly for this study, WRF-Hydro is run in coupled mode with WRF to permit the feedback of land surface fluxes of energy and moisture to the atmosphere, which impacts the simulated precipitation fields (Arnault et al. 2016, Larsen et al. 2016, Senatore et al. 2015, Koster et al. 2004). Hence, the coupled WRF/WRF-Hydro configuration differs from the standalone WRF configuration by its (i) lateral distribution of surface runoff and (ii) feedback of surface fluxes to the atmosphere.

The more physically-realistic conceptualization of terrestrial hydrologic processes in WRF-Hydro compared to the simple vertical column models used in Noah-MP in standalone WRF, along with the coupled land-atmosphere closure of the water and energy balance at each time step (15 minutes for our study), is expected to improve the accuracy of precipitation fields simulated in coupled mode.

[Figure]

Figure 1. Overland flow routing schematic from Gochis et al. (2013)

To diagnose the processes controlling the improved precipitation fields from the coupled model, we conduct two additional analyses: **(1) verification of soil moisture propagation due to lateral flow from WRF-Hydro**, and **(2) comparison of simulated surface energy balance (SEB) and planetary boundary layer (PBL) heights at the four stations considered in the study.**

**(1) Soil Moisture Propagation with Lateral Flow**

For a specific area, an individual sensor dedicated to soil moisture measurement fails to capture any change during a short time span in the order of days. The Soil Moisture Operational Products System (SMOPS), provided by the National Oceanic and Atmospheric Administration (NOAA), merges soil moisture retrievals from multi-satellites/sensors to generate a global product at higher spatial and temporal coverage (Liu et al. 2016). Relevant to the current study period, SMOPS now incorporates near-real time SMAP data and includes soil moisture retrievals from the GPM Microwave Imager (GMI). The 6-hourly product mapped at 0.25° x 0.25° spatial resolution is used here to assess the accuracy of the simulated soil moisture.

A comparison of soil moisture evolution at the upstream and downstream of a wadi within the study domain is expected to verify whether soil moisture transport occurs over the storm timescale. A wadi within the coverage of the Saih Al Salem station (24 49 39 N, 55 18 43 E) was selected to conduct this test. Fig. 2 shows the time series of simulated soil moisture from WRF/WRF-Hydro at two locations upstream and downstream of the wadi. SMOPS retrievals are overlaid as data points, along with the hyetograph recorded at the corresponding Saih Al Salem station at the top. Given the short distance (less than 1km) separating the two locations, a lag time of less than 1 hour is observed between the two soil moisture patterns. The first rain of approximately 22 mm at 22 Z 08/03/16 triggers an immediate increase in soil moisture from 0.18 to 0.25 $m^3/m^3$. The subsequent rainfall then elevates the moisture further to around 0.34 $m^3/m^3$, with a slight increase in the peak of downstream soil moisture compared to that of the upstream. However, at 18Z 09/03/16 the downstream soil moisture rises again to a sustained peak at around 0.32 $m^3/m^3$, while the upstream soil moisture continues to dissipate through infiltration and evaporation. In the absence of additional rainfall, this sustained peak in downstream soil moisture is the result of lateral surface flow from the upstream which is resolved by WRF-Hydro and fed back to the soil moisture fields. Despite the SMOPS data gaps during the event, the merged retrievals consistently increase during the event with reasonable accuracy compared to the simulated soil moisture fields.

[Figure]

**Figure 2.** Time series of simulated soil moisture from WRF/WRF-Hydro at the wadi upstream and downstream locations, along with collocated SMOPS retrievals. Hyetograph recorded at the Saih Al Salem station is shown on top.

**(2) SEB Analysis:**

Following the methodology used by Niu et al. (2011) to evaluate the performance of the Noah-MP hydrological processes at the local scale, the surface energy balance is investigated as follows:

$$\text{SW net} + \text{LW net} + (\text{Qh} + \text{Qe} + \text{Qg}) \pm \text{RES} = 0$$

, where the net shortwave (SW net) and longwave radiation (LW net) are given as the sum of the positive (outward) component and the negative (downward) components. The Qs, Ql, Qg and RES terms represent the sensible, latent, ground, and residual heat fluxes of the energy balance, respectively. The residual term arises from processes not applicable to the study domain, including energy consumed by snowmelt and rain freezing at the surface.

The Bowen ratio (β), initially proposed by Bowen (1926), describes the contributions of latent (Ql) versus sensible heat (Qs) to net radiation and can be expressed as:

$$\beta = \frac{Qs}{Ql}$$

The physical significance of the Bowen ratio is that it gives an indication of the relative partitioning of net radiation in a region.

Fig. 3 shows the SEB time series and the time-average Bowen ratio for each of the four stations: Abu Dhabi, Al Ain, Jabal Hafeet, and Jabal Mebreh.

[Figure]

**Figure 3.** Surface energy budget time series from WRF and WRF/WRF-Hydro simulations at each of the four stations and their corresponding Bowen ratios (β).

The major differences between the SEB from both model configurations are shown midway through the simulation period between 06 and 18 Z. The coupled WRF/WRF-Hydro simulation shows higher (and lower) latent (and sensible) heat fluxes, as well as slightly higher net shortwave radiation, compared to the standalone WRF simulation. The coupled model is also associated with lower Bowen ratios compared to standalone WRF. The results are in line with the soil moisture-rainfall feedback mechanisms explained by Eltahir (1998). An increase in water content of the top soil layer decreases both the surface albedo and the Bowen ratio. A lower surface albedo dictates more absorbance of net radiation, while lower Bowen ratios are a result of higher water vapor content in the boundary layer and more downwards flux of terrestrial radiation at the surface due to the water vapor greenhouse effect. This dual effect amounts to a larger total flux of heat from the surface into the boundary layer.

Furthermore, the cooling of surface temperature accompanied by the moisture should be associated with a reduced sensible heat flux and a smaller PBL height. Fig. 4 shows the PBL heights from both simulations with larger collapses resolved from the coupled model. According to Seidel et al. (2010), PBL heights can be inferred from radiosonde data (**Fig. 2 in manuscript**), particularly based on methods determining maximum or minimum vertical gradients of relative humidity or specific humidity. Such methods yield better agreement compared to those relying on locations of elevated temperature inversions or mixing height. Hence, using the Abu Dhabi radiosonde profile (**Fig. 2 in manuscript**), and based on the gradient approach, the PBL height can be estimated to be in the range of 90 – 200 m at 12 Z, which is closer to that simulated from the coupled WRF/WRF-Hydro (190 m) compared to standalone WRF (750 m).

The timings of the reduced PBL heights in Fig. 4 coincide with those of the SEB discrepancies in Fig. 3 between 06 Z and 18 Z, which corroborates the chain of events diagnosed thus far. According to Zheng and Eltahir (1998), the increase of the boundary layer moist static energy is expected to result in additional rainfall from the increase of local convection.

[Figure]

**Figure 4.** Planetary Boundary Layer (PBL) heights from WRF and WRF/WRF-Hydro simulations at each of the four stations

**Author's Changes in Manuscript:**

- 1.32 – 1.35: Additions to abstract
- 6.32 to 7.13: Statements added
- New sections 4.2.2 and 4.2.3 are added on Pages 14 and 15, respectively
- With their consent, two co-authors (Xiwu Zhan and Jicheng Liu) were added for providing the NOAA SMOPS data and advising on its use for this revision
- Figure 12 and 13 added to manuscript

**Minor and technical comments:**

Abstract: Use the degree sign instead of "o"
All degree symbols are corrected.

2.7: rain gauges → rain gauge
2.12: Corrected

3.32 And title: hyper arid: hyphenated or one word
All hyper-arid usages are now hyphenated.

4.9: rephrase to "The UAE (22 to 27N) . . . "
4.23: rephrased

4.22: rephrase to "receives more rainfall compared to the country's 100 mm annual average"
5.5: rephrased

5.28: clarify to what "Such frequencies" refers – L band?
6.13: Frequency range of 6.93 – 89 GHz is now specified

7.22: "wadis" lowercase
8.15: corrected

8.30: does "analysed products" refer to ones interpolating station observations, or those from satellite retrievals or climate models?
9.20: We are referring to the remote sensing products (satellite retrievals and merged products). This is now explicitly stated.

9.10: boarder → border
9.25: corrected

13.12: I am not certain that WRF-Hydro includes lateral flow processes that would transport soil moisture. Please verify and give more details. Whether this transport happens in reality over the storm timescale would also need to be verified, which fits in with the authors' call for a soil moisture measurement network.

In our response to C1 and the additional section 4.2.2 on Page 14, we explain that indeed soil moisture propagation is impacted by the lateral flow processes resolved by WRF-Hydro.

Table 2 and 4: Specify whether the GPM and model averages are collocated with the stations.
The statistical measures in Table 2 and 4 are obtained from collocated sites between the datasets. Both table captions are now revised accordingly.

Table 3: Specify the time interval and number of data points used for the comparison between products.
Time intervals and sample sizes (n) are now specified in the captions of both Tables 3 and 5.

**Author's References**

Arnault, J*., et al.* 2016. Role of runoff–infiltration partitioning and resolved overland flow on land–atmosphere feedbacks: A case study with the WRF-Hydro coupled modeling system for west africa. *Journal of Hydrometeorology,* 17(5), 1489-1516.

Bowen, I. S. 1926. The ratio of heat losses by conduction and by evaporation from any water surface. *Physical review,* 27(6), 779.

Ek, M*., et al.* 2003. Implementation of Noah land surface model advances in the National Centers for Environmental Prediction operational mesoscale Eta model. *Journal of Geophysical Research: Atmospheres,* 108(D22).

Eltahir, E. A. 1998. A soil moisture–rainfall feedback mechanism: 1. Theory and observations. *Water Resources Research,* 34(4), 765-776.

Gochis, D., Yu, W. and Yates, D. 2013. The WRF-Hydro model technical description and user's guide, version 1.0. *NCAR Tech. Doc*.

Koster, R. D*., et al.* 2004. Regions of strong coupling between soil moisture and precipitation. *Science,* 305(5687), 1138-1140.

Larsen, M. A*., et al.* 2016. Local control on precipitation in a fully coupled climate-hydrology model. *Scientific reports,* 6, 22927.

Liu, J*., et al.*, NOAA Soil Moisture Operational Product System (SMOPS) and its validations. ed. *2016 IEEE International Geoscience and Remote Sensing Symposium (IGARSS)*, 2016, 3477-3480.

Niu, G. Y*., et al.* 2011. The community Noah land surface model with multiparameterization options (Noah-MP): 1. Model description and evaluation with local-scale measurements. *Journal of Geophysical Research: Atmospheres,* 116(D12).

Seidel, D. J., Ao, C. O. and Li, K. 2010. Estimating climatological planetary boundary layer heights from radiosonde observations: Comparison of methods and uncertainty analysis. *Journal of Geophysical Research: Atmospheres,* 115(D16).

Senatore, A*., et al.* 2015. Fully coupled atmosphere-hydrology simulations for the central Mediterranean: Impact of enhanced hydrological parameterization for short and long time scales. *Journal of Advances in Modeling Earth Systems,* 7(4), 1693-1715.

Zheng, X. and Eltahir, E. A. 1998. A soil moisture–rainfall feedback mechanism: 2. Numerical experiments. *Water Resources Research,* 34(4), 777-785.

**Response to Reviewer 2**

We thank Reviewer 2 for his/her insightful comments and suggestions. Below is our point-by-point responses to the provided comments. A marked-up version of the revised manuscript is also appended.

Reviewer Summary: This manuscript presents an assessment of modelled rainfall patterns and amounts for an extreme rainfall event in UAE derived from two modelling systems, namely, the standalone WRF and the coupled WRF/Hydro system. The evaluation of model results is based on a comparison with weather stations' data (i.e. gauge rainfall data, temperature, radiation) and satellite products (i.e. the Global Precipitation Measurement (GPM) rainfall, the MODIS cloud fraction, and ASMR2 soil moisture). In the manuscript, analysed variables are limited to these hydrometeorological variables, i.e. precipitation, cloud cover, global radiation, air temperature, and soil moisture. Statistical output of the evaluation shows that the coupled WRF/Hydro is better than the standalone WRF. However, no further effort is made to diagnose the processes and mechanisms controlling the water cycle that can be better captured by the coupled WRF/Hydro system than the standalone WRF. Thus I recommend that revision should be made for the following key points:

In line with the reviewer's suggestion, we have carried out two additional analyses to diagnose the processes and mechanisms controlling the improved precipitation fields in the coupled model. First, we analyze the impact of lateral flow on the propagation of soil moisture captured by WRF-Hydro during the event. Second, we compare the simulated surface energy balance (SEB) and planetary boundary layer (PBL) heights at the four stations considered in the study. The outcomes of the additional analysis corroborate our initial findings and demonstrate how soil moisture and atmospheric water vapor alter the distribution of fluxes and affect the PBL height. We added new figures and the supporting analyses (included in our following responses).

**Comment 1 (C1)**: Literature review of the manuscript stated that numerous studies in the past have already shown the advantages of the coupled WRF/Hydro over the standalone WRF. If this study is a same kind but just a case study for another geographical location, what would be its unique contributions to knowledge?

**Author's response to C1**: Hydrological processes in hyper-arid regions are different from those in mid latitude regions. In desert regions, high soil porosity and hydraulic conductivity of the prevailing sandy soil implies rapid infiltration and runoff drainage. This suggests that the impact on latent heat and, therefore, on the surface radiation budget would be minimal. This study, with the additional analyses during the revision, demonstrates that even in desert regions the surface feedback to the atmosphere is still considerable and important to account for. We state on Page 4, Line 11 that "such coupling has never been assessed in hyper-arid environments like the one observed in the UAE, where hydrological and atmospheric processes are specific and different from other study domains where similar coupling was evaluated".

We explain that the gentle topography in the study area, with a slope favoring water drainage from the east to the west, does not drain water rapidly. Furthermore, precipitation largely contributes to soil physical crust formation in desert environments as shown by Fang et al. (2007). Precipitation compacts fine particulate and fills the porosities of the top soil layer, forming a hard shell. Dust is also washed out of the atmosphere by precipitation over desert environments which increases amounts of finer particulate at the surface layer to further accelerate crust formation process. This translates to less vertical infiltration

and more lateral flow processes. These mechanisms are specific to arid regions and corroborate the importance of accounting for lateral flow and surface feedback in the coupled WRF/WRF-Hydro model to correctly capture the atmospheric and hydrological process.

In their Global Land Atmosphere Coupling Experiment (GLACE), Koster et al. (2004) identified regional hot spots for coupling strengths, including moderate coupling strengths over the Arabian Peninsula. A follow-on study from Seneviratne et al. (2006) followed the same methodology as GLACE, but with higher resolution model runs, found high coupling strengths over Europe which were not previously reported by GLACE. Consequently, the present study represents the first local assessment of coupling over the UAE for short-term (48 hours) and high-resolution (100-meter) prediction of an extreme event, and quantifies the added value of coupling for the accuracy of precipitation forecasts.

**Author's Changes in manuscript**: Additions on Page 3, Lines 33 – Page 4, Line 7.

**Comment 2 (C2)**: As claimed in the manuscript, the main objective of the study is to investigate the added value of coupled land surface-atmospheric modeling (WRF-Hydro) over the hyper-arid environment of the UAE. In fact, the coupled WRF-Hydro system captures the dynamics of the water and energy cycles, linking the upper atmosphere to the unsaturated and saturated zones on the land surface. In order to take the full advantage of the WRF-Hydro system, diagnoses of the feedback processes/mechanisms controlling the regional scale water cycle (e.g. runoff, penetration, evaporative fraction, water vapour flux) should be conducted. Such diagnoses may lead to valuable generic outcome that could benefit the research community. In fact, the discussion in the manuscript has cited many publications for such processes/mechanisms for the purpose of interpreting the modelled output, but none of these has been further diagnosed in this study. It is strongly recommended that these diagnoses should be explored.

**Author's response to C2**: We fully agree with the reviewer on the need to diagnose the mentioned processes in more depth. The scarcity of in situ data for runoff and flux measurements is a major challenge in the study region (Ghebreyesus et al. 2016, Wehbe et al. 2017). Nevertheless, we have carried out additional analyses based on the comparison of simulated surface energy balance (SEB) and planetary boundary layer (PBL) heights between the two models at the four stations considered in the study.

Following the methodology used by Niu et al. (2011) to evaluate the performance of the Noah-MP hydrological processes at the local scale, the surface energy balance is investigated as follows:

$$SW\,net\,+\,LW\,net\,+\,(Qh\,+\,Qe\,+\,Qg) \pm RES\,=\,0$$

, where the net shortwave (SW net) and longwave radiation (LW net) are given as the sum of the positive (outward) component and the negative (downward) components. The Qs, Ql, Qg and RES terms represent the sensible, latent, ground, and residual heat fluxes of the energy balance, respectively. The residual term arises from processes not applicable to the study domain, including energy consumed by snowmelt and rain freezing at the surface.

The Bowen ratio ($\beta$), initially proposed by Bowen (1926), gives an indication of the relative partitioning of net radiation in a region and can be expressed as:

$$\beta = \frac{Qs}{Ql}$$

Fig. 1 shows the SEB time series and the time-averaged Bowen ratio at each of the four studied stations.

[Figure]

**Figure 1.** Surface energy budget time series from WRF and WRF/WRF-Hydro simulations at each of the four stations and their corresponding Bowen ratios (β).

The major differences between the SEB from both model configurations are shown midway through the simulation period between 06 and 18 Z. The coupled WRF/WRF-Hydro simulation shows higher (and lower) latent (and sensible) heat fluxes, as well as slightly higher net shortwave radiation, compared to the standalone WRF simulation. The coupled model is also associated with lower Bowen ratios compared to standalone WRF. The results are in line with the soil moisture-rainfall feedback mechanisms explained by Eltahir (1998). An increase in water content of the top soil layer decreases both the surface albedo and the Bowen ratio. A lower surface albedo dictates more absorbance of net radiation, while lower Bowen ratios are a result of higher water vapor content in the boundary layer and more downwards flux of terrestrial radiation at the surface due to the water vapor greenhouse effect. This dual effect amounts to a larger total flux of heat from the surface into the boundary layer.

Furthermore, the cooling of surface temperature accompanied by the moisture should be associated with a reduced sensible heat flux and a smaller PBL height. Fig. 2 shows the PBL heights from both simulations with larger collapses resolved from the coupled model. According to Seidel et al. (2010), PBL heights can be inferred from radiosonde data (**Fig. 2 in manuscript**), particularly based on methods determining maximum or minimum vertical gradients of relative humidity or specific humidity. Such methods yield better agreement compared to those relying on locations of elevated temperature inversions or mixing height. Hence, using the Abu Dhabi radiosonde profile (**Fig. 2 in manuscript**), and based on the gradient approach, the PBL height can be estimated to be in the range of 90 – 200 m at 12 Z, which is closer to that simulated from the coupled WRF/WRF-Hydro (190 m) compared to standalone WRF (750 m).

The timings of the reduced PBL heights in Fig. 2 coincide with those of the SEB discrepancies in Fig. 1 between 06 Z and 18 Z, which corroborates the occurrence of the chain of events considered thus far. According to Zheng and Eltahir (1998), the increase of the boundary layer moist static energy is expected to result in additional rainfall from the increase of local convection.

[Figure]

**Figure 2.** Planetary Boundary Layer (PBL) heights from WRF and WRF/WRF-Hydro simulations at each of the four stations

**Author's Changes in manuscript**:
- Statements added on Page 6, Line 32 – Page 7, Line 13
- New section 4.2.3 and statements added on Page 15, Lines 6 – 17
- Figure 13 added to manuscript (Page 39)

**Comment 3 (C3)**: Several speculative arguments (e.g. lines 31-33 of p.10 about the processes linking rainfall to soil moisture and to 2m air temperature, lines 5-6 of p.11 about the effect of soil moisture on surface emissivity/temperature, lines 11-15 of p. 12 about resolved scale vs subgrid scale cumulus, lines 13-15 of p.12 about underestimation of cloud by MODIS, and lines 19-20 of p. 12 about spin-up time) may be further analyzed in order to show in-depth processes.

**Author's response to C3**: Following Comment 2, we thank the reviewer and fully agree on the need for further analyses of in-depth processes, particularly the hydrological processes. In the below, we pursue the verification of soil moisture propagation due to lateral flow resolved by WRF-Hydro.

For a specific area, an individual sensor dedicated to soil moisture measurement fails to capture any change during a short time span in the order of days. The Soil Moisture Operational Products System (SMOPS), provided by the National Oceanic and Atmospheric Administration (NOAA), merges soil moisture retrievals from multi-satellites/sensors to generate a global product at higher spatial and temporal coverage (Liu et al. 2016). Relevant to the current study period, SMOPS now incorporates near-real time SMAP data and includes soil moisture retrievals from the GPM Microwave Imager (GMI). The 6-hourly product mapped at 0.25° x 0.25° spatial resolution is used here to assess the accuracy of the simulated soil moisture.

A comparison of soil moisture evolution at the upstream and downstream of a wadi within the study domain is expected to verify whether soil moisture transport occurs over the storm timescale. A wadi within the coverage of the Saih Al Salem station (24 49 39 N, 55 18 43 E) was selected to conduct this test. Fig. 3 shows the time series of simulated soil moisture from WRF/WRF-Hydro at two locations upstream and downstream of the wadi. SMOPS retrievals are overlaid as data points, along with the hyetograph recorded at the corresponding Saih Al Salem station at the top. Given the short distance (less than 1km) separating the two locations, a lag time of less than 1 hour is observed between the two soil moisture patterns. The first rain of approximately 22 mm at 22 Z 08/03/16 triggers an immediate increase in soil moisture from 0.18 to 0.25 $m^3/m^3$. The subsequent rainfall then elevates the moisture further to around 0.34 $m^3/m^3$, with a slight increase in the peak of downstream soil moisture compared to that of the upstream. However, at 18Z 09/03/16 the downstream soil moisture rises again to a sustained peak at around 0.32 $m^3/m^3$, while the upstream soil moisture continues to dissipate through infiltration and evaporation. In the absence of additional rainfall, this sustained peak in downstream soil moisture is the result of lateral surface flow from the upstream which is resolved by WRF-Hydro and fed back to the soil moisture fields. Despite the SMOPS data gaps during the event, the merged retrievals consistently increase during the event with reasonable accuracy compared to the simulated soil moisture fields.

[Figure]

**Figure 3.** Time series of simulated soil moisture from WRF/WRF-Hydro at the wadi upstream and downstream locations, along with collocated SMOPS retrievals. Hyetograph recorded at the Saih Al Salem station is shown on top.

**Author's Changes in manuscript**:
- Additions to abstract on Page 1, Lines 32 – 35.
- New section 4.2.2 added on Page 14, Lines 12 – 34
- Figure 12 added to manuscript (Page 38)
- With their consent, two co-authors (Xiwu Zhan and Jicheng Liu) were added for providing the NOAA SMOPS data and advising on its use for this revision

**Comment 4 (C4)**: Figure 3 (c) & (d) and Figure 10's soil moisture plots from WRF all have shown weird stripe structure of modelled accumulated rainfall and soil moisture, respectively. This adds doubts to model settings or post-processing and must be investigated thoroughly and the reasons should be fully explained. Once the errors are identified, all analyses should be re-done and all results should be updated.

**Author's response to C4**: We thank the reviewer for capturing this anomaly in the mentioned figures. We revisited the raw WRF output files and found the problem arising from the post-processing, particularly the interpolation fields. The Gaussian-weighted interpolation routine was initially used during the post-processing. We have repeated the post-processing for both figures using the Cressman-weighted interpolation, which conducts successive corrections using a decreasing radius of influence. This method required more computational time, but the results retained more of the mesoscale structure. New figures are added in the revised manuscript.

**Author's References**

Bowen, I. S. 1926. The ratio of heat losses by conduction and by evaporation from any water surface. *Physical review,* 27(6), 779.

Eltahir, E. A. 1998. A soil moisture–rainfall feedback mechanism: 1. Theory and observations. *Water Resources Research,* 34(4), 765-776.

Fang, H. Y.*, et al.* 2007. Mechanism of formation of physical soil crust in desert soils treated with straw checkerboards. *Soil and Tillage Research,* 93(1), 222-230.

Ghebreyesus, D.*, et al.* 2016. A multi-satellite approach for water storage monitoring in an arid watershed. *Geosciences,* 6(3), 33.

Koster, R. D.*, et al.* 2004. Regions of strong coupling between soil moisture and precipitation. *Science,* 305(5687), 1138-1140.

Liu, J.*, et al.*, NOAA Soil Moisture Operational Product System (SMOPS) and its validations. ed. *2016 IEEE International Geoscience and Remote Sensing Symposium (IGARSS)*, 2016, 3477-3480.

Niu, G. Y.*, et al.* 2011. The community Noah land surface model with multiparameterization options (Noah-MP): 1. Model description and evaluation with local-scale measurements. *Journal of Geophysical Research: Atmospheres,* 116(D12).

Seidel, D. J., Ao, C. O. and Li, K. 2010. Estimating climatological planetary boundary layer heights from radiosonde observations: Comparison of methods and uncertainty analysis. *Journal of Geophysical Research: Atmospheres,* 115(D16).

Seneviratne, S. I.*, et al.* 2006. Land–atmosphere coupling and climate change in Europe. *Nature,* 443(7108), 205.

Wehbe, Y.*, et al.* 2017. Assessment of the consistency among global precipitation products over the United Arab Emirates. *Journal of Hydrology: Regional Studies,* 12, 122-135.

Zheng, X. and Eltahir, E. A. 1998. A soil moisture–rainfall feedback mechanism: 2. Numerical experiments. *Water Resources Research,* 34(4), 777-785.

**Analysis of an Extreme Weather Event in a Hyper--Arid Region Using WRF-Hydro Coupling, Station, and Satellite data**

Youssef Wehbe[1,2], Marouane Temimi[1], Michael Weston[1], Naira Chaouch[13], Oliver Branch[34], Thomas Schwitalla[34], Volker Wulfmeyer[34], Xiwu Zhan[5], Jicheng Liu[5,6], and Abdulla Al Mandous[2]

[1] Department of Civil Infrastructure and Environmental Engineering, Khalifa University of Science and Technology, Masdar City, P.O. Box 54224 Abu Dhabi, United Arab Emirates
[2] National Center of Meteorology (NCM), P.O. Box 4815, Abu Dhabi, United Arab Emirates
[3] NOAA CREST Institute/City University of New York, New York, NY, United States
[43] Institute of Physics and Meteorology, University of Hohenheim, Garbenstraße 30, D-70599 Stuttgart, Germany
[5] NOAA-NESDIS Center for Satellite Applications and Research (STAR) NOAA
[6] ESSIC/CICS, University of Maryland College Park, College Park 20740, MD, USA

*Correspondence to*: Youssef Wehbe (ywehbe@ncms.ae)

**Abstract.** This study investigates an extreme weather event that impacted the United Arab Emirates (UAE) in March 2016 using the Weather Research and Forecasting (WRF) model version 3.7.1 coupled with its hydrological modeling extension package (Hydro). Six-hourly forecasted forcing records at $0.5°\text{o}$ spatial resolution, obtained from the NCEP Global Forecast System (GFS), are used to drive the three nested downscaling domains of both standalone WRF and coupled WRF/WRF-Hydro configurations for the recent flood-triggering storm. Ground and satellite observations over the UAE are employed to validate the model results. The model performance was assessed using pPrecipitation from GPM (30-minute, 0.1° product), soil moisture from AMSR2 (daily, 0.1° product) and NOAA SMOPS global product (6-hourly, 0.25° product), and cloud fraction retrievals from GPM (30-minute, 0.1o product), AMSR2 (daily, 0.1o product), and MODIS (daily, 5 km product), respectively, are used to assess the model output. 
[revised manuscript text omitted]